# Accelerated viral dynamics in bat cell lines, with implications for zoonotic emergence

Cara E Brook[1,2]*, Mike Boots[1], Kartik Chandran[3], Andrew P Dobson[2], Christian Drosten[4], Andrea L Graham[2], Bryan T Grenfell[2,5], Marcel A Müller[4,6], Melinda Ng[3], Lin-Fa Wang[7], Anieke van Leeuwen[2,8]

[1]Department of Integrative Biology, University of California, Berkeley, Berkeley, United States; [2]Department of Ecology and Evolutionary Biology, Princeton University, Princeton, United States; [3]Department of Microbiology and Immunology, Albert Einstein College of Medicine, Bronx, United States; [4]Institute of Virology, Charité-Universitätsmedizin Berlin, corporate member of Freie Universität Berlin, Humboldt-Universität zu Berlin, and Berlin Institute of Health, Berlin, Germany; [5]Fogarty International Center, National Institutes of Health, Bethesda, United States; [6]Martsinovsky Institute of Medical Parasitology, Tropical and Vector Borne Diseases, Sechenov University, Moscow, Russian Federation; [7]Emerging Infectious Diseases Program, Duke-National University of Singapore Medical School, Singapore, Singapore; [8]Royal Netherlands Institute for Sea Research, Department of Coastal Systems, and Utrecht University, Den Burg, Netherlands

*For correspondence:
cbrook@berkeley.edu

Competing interests: The authors declare that no competing interests exist.

**Abstract** Bats host virulent zoonotic viruses without experiencing disease. A mechanistic understanding of the impact of bats' virus hosting capacities, including uniquely constitutive immune pathways, on cellular-scale viral dynamics is needed to elucidate zoonotic emergence. We carried out virus infectivity assays on bat cell lines expressing induced and constitutive immune phenotypes, then developed a theoretical model of our *in vitro* system, which we fit to empirical data. Best fit models recapitulated expected immune phenotypes for representative cell lines, supporting robust antiviral defenses in bat cells that correlated with higher estimates for within-host viral propagation rates. In general, heightened immune responses limit pathogen-induced cellular morbidity, which can facilitate the establishment of rapidly-propagating persistent infections within-host. Rapidly-transmitting viruses that have evolved with bat immune systems will likely cause enhanced virulence following emergence into secondary hosts with immune systems that diverge from those unique to bats.

## Introduction

Bats have received much attention in recent years for their role as reservoir hosts for emerging viral zoonoses, including rabies and related lyssaviruses, Hendra and Nipah henipaviruses, Ebola and Marburg filoviruses, and SARS coronavirus (*Calisher et al., 2006*; *Wang and Anderson, 2019*). In most non-Chiropteran mammals, henipaviruses, filoviruses, and coronaviruses induce substantial morbidity and mortality, display short durations of infection, and elicit robust, long-term immunity in hosts surviving infection (*Nicholls et al., 2003*; *Hooper et al., 2001*; *Mahanty and Bray, 2004*). Bats, by contrast, demonstrate no obvious disease symptoms upon infection with pathogens that are highly virulent in non-volant mammals (*Schountz et al., 2017*) but may, instead, support viruses as long-term persistent infections, rather than transient, immunizing pathologies (*Plowright et al., 2016*).

**eLife digest** Bats can carry viruses that are deadly to other mammals without themselves showing serious symptoms. In fact, bats are natural reservoirs for viruses that have some of the highest fatality rates of any viruses that people acquire from wild animals – including rabies, Ebola and the SARS coronavirus.

Bats have a suite of antiviral defenses that keep the amount of virus in check. For example, some bats have an antiviral immune response called the interferon pathway perpetually switched on. In most other mammals, having such a hyper-vigilant immune response would cause harmful inflammation. Bats, however, have adapted anti-inflammatory traits that protect them from such harm, include the loss of certain genes that normally promote inflammation. However, no one has previously explored how these unique antiviral defenses of bats impact the viruses themselves.

Now, Brook et al. have studied this exact question using bat cells grown in the laboratory. The experiments made use of cells from one bat species – the black flying fox – in which the interferon pathway is always on, and another – the Egyptian fruit bat – in which this pathway is only activated during an infection. The bat cells were infected with three different viruses, and then Brook et al. observed how the interferon pathway helped keep the infections in check, before creating a computer model of this response.

The experiments and model helped reveal that the bats' defenses may have a potential downside for other animals, including humans. In both bat species, the strongest antiviral responses were countered by the virus spreading more quickly from cell to cell. This suggests that bat immune defenses may drive the evolution of faster transmitting viruses, and while bats are well protected from the harmful effects of their own prolific viruses, other creatures like humans are not.

The findings may help to explain why bats are often the source for viruses that are deadly in humans. Learning more about bats' antiviral defenses and how they drive virus evolution may help scientists develop better ways to predict, prevent or limit the spread of viruses from bats to humans. More studies are needed in bats to help these efforts. In the meantime, the experiments highlight the importance of warning people to avoid direct contact with wild bats.

Recent research advances are beginning to shed light on the molecular mechanisms by which bats avoid pathology from these otherwise virulent pathogens (*Brook and Dobson, 2015*). Bats leverage a suite of species-specific mechanisms to limit viral load, which include host receptor sequence incompatibilities for some bat-virus combinations (*Ng et al., 2015*; *Takadate et al., 2020*) and constitutive expression of the antiviral cytokine, IFN-α, for others (*Zhou et al., 2016*). Typically, the presence of viral RNA or DNA in the cytoplasm of mammalian cells will induce secretion of type I interferon proteins (IFN-α and IFN-β), which promote expression and translation of interferon-stimulated genes (ISGs) in neighboring cells and render them effectively antiviral (*Stetson and Medzhitov, 2006*). In some bat cells, the transcriptomic blueprints for this IFN response are expressed constitutively, even in the absence of stimulation by viral RNA or DNA (*Zhou et al., 2016*). In non-flying mammals, constitutive IFN expression would likely elicit widespread inflammation and concomitant immunopathology upon viral infection, but bats support unique adaptations to combat inflammation (*Zhang et al., 2013*; *Ahn et al., 2019*; *Xie et al., 2018*; *Pavlovich et al., 2018*) that may have evolved to mitigate metabolic damage induced during flight (*Kacprzyk et al., 2017*). The extent to which constitutive IFN-α expression signifies constitutive antiviral defense in the form of functional IFN-α protein remains unresolved. In bat cells constitutively expressing IFN-α, some protein-stimulated, downstream ISGs appear to be also constitutively expressed, but additional ISG induction is nonetheless possible following viral challenge and stimulation of IFN-β (*Zhou et al., 2016*; *Xie et al., 2018*). Despite recent advances in molecular understanding of bat viral tolerance, the consequences of this unique bat immunity on within-host virus dynamics—and its implications for understanding zoonotic emergence—have yet to be elucidated.

The field of 'virus dynamics' was first developed to describe the mechanistic underpinnings of long-term patterns of steady-state viral load exhibited by patients in chronic phase infections with HIV, who appeared to produce and clear virus at equivalent rates (*Nowak and May, 2000*; *Ho et al., 1995*). Models of simple target cell depletion, in which viral load is dictated by a bottom-

up resource supply of infection-susceptible host cells, were first developed for HIV (*Perelson, 2002*) but have since been applied to other chronic infections, including hepatitis-C virus (*Neumann et al., 1998*), hepatitis-B virus (*Nowak et al., 1996*) and cytomegalovirus (*Emery et al., 1999*). Recent work has adopted similar techniques to model the within-host dynamics of acute infections, such as influenza A and measles, inspiring debate over the extent to which explicit modeling of top-down immune control can improve inference beyond the basic resource limitation assumptions of the target cell model (*Baccam et al., 2006*; *Pawelek et al., 2012*; *Saenz et al., 2010*; *Morris et al., 2018*).

To investigate the impact of unique bat immune processes on *in vitro* viral kinetics, we first undertook a series of virus infection experiments on bat cell lines expressing divergent interferon phenotypes, then developed a theoretical model elucidating the dynamics of within-host viral spread. We evaluated our theoretical model analytically independent of the data, then fit the model to data recovered from *in vitro* experimental trials in order to estimate rates of within-host virus transmission and cellular progression to antiviral status under diverse assumptions of absent, induced, and constitutive immunity. Finally, we confirmed our findings in spatially-explicit stochastic simulations of fitted time series from our mean field model. We hypothesized that top-down immune processes would overrule classical resource-limitation in bat cell lines described as constitutively antiviral in the literature, offering a testable prediction for models fit to empirical data. We further predicted that the most robust antiviral responses would be associated with the most rapid within-host virus propagation rates but also protect cells against virus-induced mortality to support the longest enduring infections in tissue culture.

## Results

### Virus infection experiments in antiviral bat cell cultures yield reduced cell mortality and elongated epidemics

We first explored the influence of innate immune phenotype on within-host viral propagation in a series of infection experiments in cell culture. We conducted plaque assays on six-well plate monolayers of three immortalized mammalian kidney cell lines: [1] Vero (African green monkey) cells, which are IFN-defective and thus limited in antiviral capacity (*Desmyter et al., 1968*); [2] RoNi/7.1 (*Rousettus aegyptiacus*) cells which demonstrate idiosyncratic induced interferon responses upon viral challenge (*Kuzmin et al., 2017*; *Arnold et al., 2018*; *Biesold et al., 2011*; *Pavlovich et al., 2018*); and [3] PaKiT01 (*Pteropus alecto*) cells which constitutively express IFN-α (*Zhou et al., 2016*; *Crameri et al., 2009*). To intensify cell line-specific differences in constitutive immunity, we carried out infectivity assays with GFP-tagged, replication-competent vesicular stomatitis Indiana viruses: rVSV-G, rVSV-EBOV, and rVSV-MARV, which have been previously described (*Miller et al., 2012*; *Wong et al., 2010*). Two of these viruses, rVSV-EBOV and rVSV-MARV, are recombinants for which cell entry is mediated by the glycoprotein of the bat-evolved filoviruses, Ebola (EBOV) and Marburg (MARV), thus allowing us to modulate the extent of structural, as well as immunological, antiviral defense at play in each infection. Previous work in this lab has demonstrated incompatibilities in the NPC1 filovirus receptor which render PaKiT01 cells refractory to infection with rVSV-MARV (Ng and Chandrab, 2018, Unpublished results), making them structurally antiviral, over and above their constitutive expression of IFN-α. All three cell lines were challenged with all three viruses at two multiplicities of infection (MOI): 0.001 and 0.0001. Between 18 and 39 trials were run at each cell-virus-MOI combination, excepting rVSV-MARV infections on PaKiT01 cells at MOI = 0.001, for which only eight trials were run (see Materials and methods; *Figure 1—figure supplements 1–3*, *Supplementary file 1*).

Because plaque assays restrict viral transmission neighbor-to-neighbor in two-dimensional cellular space (*Howat et al., 2006*), we were able to track the spread of GFP-expressing virus-infected cells across tissue monolayers via inverted fluorescence microscopy. For each infection trial, we monitored and re-imaged plates for up to 200 hr of observations or until total monolayer destruction, processed resulting images, and generated a time series of the proportion of infectious-cell occupied plate space across the duration of each trial (see Materials and methods). We used generalized additive models to infer the time course of all cell culture replicates and construct the multi-trial dataset to which we eventually fit our mechanistic transmission model for each cell line-virus-specific combination (*Figure 1*; *Figure 1—figure supplements 1–5*).

All three recombinant vesicular stomatitis viruses (rVSV-G, rVSV-EBOV, and rVSV-MARV) infected Vero, RoNi/7.1, and PaKiT01 tissue cultures at both focal MOIs. Post-invasion, virus spread rapidly across most cell monolayers, resulting in virus-induced epidemic extinction. Epidemics were less severe in bat cell cultures, especially when infected with the recombinant filoviruses, rVSV-EBOV and rVSV-MARV. Monolayer destruction was avoided in the case of rVSV-EBOV and rVSV-MARV infections on PaKiT01 cells: in the former, persistent viral infection was maintained throughout the 200 hr duration of each experiment, while, in the latter, infection was eliminated early in the time series, preserving a large proportion of live, uninfectious cells across the duration of the experiment. We assumed this pattern to be the result of immune-mediated epidemic extinction (*Figure 1*). Patterns from MOI = 0.001 were largely recapitulated at MOI = 0.0001, though at somewhat reduced total proportions (*Figure 1—figure supplement 5*).

## A theoretical model fit to *in vitro* data recapitulates expected immune phenotypes for bat cells

We next developed a within-host model to fit to these data to elucidate the effects of induced and constitutive immunity on the dynamics of viral spread in host tissue (*Figure 1*). The compartmental within-host system mimicked our two-dimensional cell culture monolayer, with cells occupying five distinct infection states: susceptible (S), antiviral (A), exposed (E), infectious (I), and dead (D). We modeled exposed cells as infected but not yet infectious, capturing the 'eclipse phase' of viral integration into a host cell which precedes viral replication. Antiviral cells were immune to viral infection, in accordance with the 'antiviral state' induced from interferon stimulation of ISGs in tissues adjacent to infection (*Stetson and Medzhitov, 2006*). Because we aimed to translate available data into modeled processes, we did not explicitly model interferon dynamics but instead scaled the rate of cell progression from susceptible to antiviral ($\rho$) by the proportion of exposed cells (globally) in the system. In systems permitting constitutive immunity, a second rate of cellular acquisition of antiviral status ($\varepsilon$) additionally scaled with the global proportion of susceptible cells in the model. Compared with virus, IFN particles are small and highly diffusive, justifying this global signaling assumption at the limited spatial extent of a six-well plate and maintaining consistency with previous modeling approximations of IFN signaling in plaque assay (*Howat et al., 2006*).

To best represent our empirical monolayer system, we expressed our state variables as proportions ($P_S$, $P_A$, $P_E$, $P_I$, and $P_D$), under assumptions of frequency-dependent transmission in a well-mixed population (*Keeling and Rohani, 2008*), though note that the inclusion of $P_D$ (representing the proportion of dead space in the modeled tissue) had the functional effect of varying transmission with infectious cell density. This resulted in the following system of ordinary differential equations:

$$\frac{dP_S}{dt} = bP_D(P_S + P_A) - \beta P_S P_I - \mu P_S - \rho P_E P_S - \varepsilon P_S + c P_A \tag{1}$$

$$\frac{dP_A}{dt} = \rho P_E P_S + \varepsilon P_S - c P_A - \mu P_A \tag{2}$$

$$\frac{dP_E}{dt} = \beta P_S P_I - \sigma P_E - \mu P_E \tag{3}$$

$$\frac{dP_I}{dt} = \sigma P_E - \alpha P_I - \mu P_I \tag{4}$$

$$\frac{dP_D}{dt} = \mu(P_S + P_E + P_I + P_A) + \alpha P_I - bP_D(P_S + P_A) \tag{5}$$

We defined 'induced immunity' as complete, modeling all cells as susceptible to viral invasion at disease-free equilibrium, with defenses induced subsequent to viral exposure through the term $\rho$. By contrast, we allowed the extent of constitutive immunity to vary across the parameter range of $\varepsilon >$ 0, defining a 'constitutive' system as one containing *any* antiviral cells at disease-free equilibrium. In fitting this model to tissue culture data, we independently estimated both $\rho$ and $\varepsilon$, as well as the cell-to-cell transmission rate, $\beta$, for each cell-virus combination. Since the extent to which

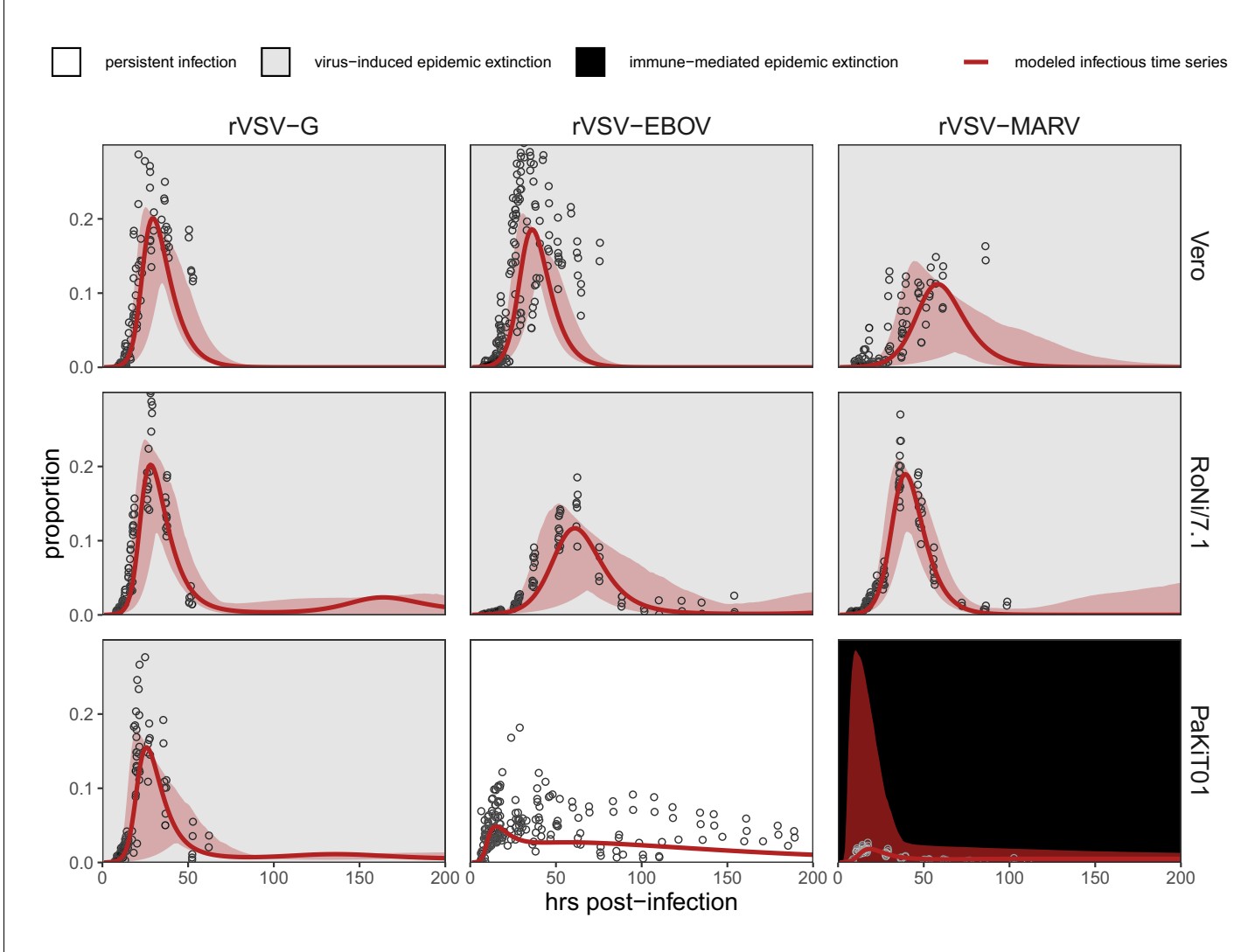

**Figure 1.** Fitted time series of infectious cell proportions from mean field model for rVSV-G, rVSV-EBOV, and rVSV-MARV infections (columns) on Vero, RoNi/7.1, and PaKiT01 cell lines (rows) at MOI = 0.001. Results are shown for the best fit immune absent model on Vero cells, induced immunity model on RoNi/7.1 cells, and constitutive (for rVSV-VSVG and rVSV-EBOV) and induced (for rVSV-MARV) immunity models on PaKiT01 cells. Raw data across all trials are shown as open circles (statistical smoothers from each trial used for fitting are available in *Figure 1—figure supplements 2–3*). Model output is shown as a solid crimson line (95% confidence intervals by standard error = red shading). Panel background corresponds to empirical outcome of the average stochastic cell culture trial (persistent infection = white; virus-induced epidemic extinction = gray; immune-mediated epidemic extinction = black). Parameter values are listed in *Table 1* and *Supplementary file 4*. Results for absent/induced/constitutive fitted models across all cell lines are shown in *Figure 1—figure supplement 4* (MOI = 0.001) and *Figure 1—figure supplement 5* (MOI = 0.0001).

The online version of this article includes the following figure supplement(s) for figure 1:

**Figure supplement 1.** Cell culture models of viral propagation.

**Figure supplement 2.** Time series data to which mean field mechanistic models were fit, across rVSV-G (left), rVSV-EBOV (middle), and rVSV-MARV (right) infections on Vero, RoNi/7.1, and PaKiT01 cell lines, at MOI = 0.001.

**Figure supplement 3.** Time series data to which mean field mechanistic models were fit, across rVSV-G (left), rVSV-EBOV (middle), and rVSV-MARV (right) infections on Vero, RoNi/7.1, and PaKiT01 cell lines, at MOI = 0.0001.

**Figure supplement 4.** Figure replicates *Figure 1* (main text) but includes all output across mean field model fits assuming (**A**) absent immunity, (**B**) induced immunity, and (**C**) constitutive immunity.

**Figure supplement 5.** Figure replicates *Figure 1—figure supplement 4* exactly but shows model fits and data for all cell-virus combinations at MOI = 0.0001.

**Figure supplement 6.** IFN gene expression in bat cells at baseline and upon viral stimulation.

**Figure supplement 7.** Curve fits to control data for standard birth ($b = .025$) and natural mortality ($\mu = \frac{1}{121}, \frac{1}{191}, \frac{1}{84}$ hours for, respectively, Vero, RoNi/7.1, and PaKiT01 cell lines) rates across all three cell lines.

constitutively-expressed IFN-α is constitutively translated into functional protein is not yet known for bat hosts (*Zhou et al., 2016*), this approach permitted our tissue culture data to drive modeling inference: even in PaKiT01 cell lines known to constitutively express IFN-α, the true constitutive extent of the system (i.e. the quantity of antiviral cells present at disease-free equilibrium) was allowed to vary through estimation of $\varepsilon$. For the purposes of model-fitting, we fixed the value of $c$, the return rate of antiviral cells to susceptible status, at 0. The small spatial scale and short time course (max 200 hours) of our experiments likely prohibited any return of antiviral cells to susceptible status in our empirical system; nonetheless, we retained the term $c$ in analytical evaluations of our model because regression from antiviral to susceptible status is possible over long time periods *in vitro* and at the scale of a complete organism (*Radke et al., 1974*; *Rasmussen and Farley, 1975*; *Samuel and Knutson, 1982*).

Before fitting to empirical time series, we undertook bifurcation analysis of our theoretical model and generated testable hypotheses on the basis of model outcomes. From our within-host model system (*Equation 1-5*), we derived the following expression for $R_0$, the pathogen basic reproduction number (*Supplementary file 2*):

$$R_0 = \frac{\beta\sigma(b-\mu)(c+\mu)}{b(\sigma+\mu)(\alpha+\mu)(c+\mu+\varepsilon)} \tag{6}$$

Pathogens can invade a host tissue culture when $R_0>1$. Rapid rates of constitutive antiviral acquisition ($\varepsilon$) will drive $R_0<1$: tissue cultures with highly constitutive antiviral immunity will be therefore resistant to virus invasion from the outset. Since, by definition, induced immunity is stimulated following initial virus invasion, the rate of induced antiviral acquisition (ρ) is not incorporated into the equation for $R_0$; while induced immune processes can control virus after initial invasion, they cannot prevent it from occurring to begin with. In cases of fully induced or absent immunity ($\varepsilon=0$), the $R_0$ equation thus reduces to a form typical of the classic SEIR model:

$$R_0 = \frac{\beta\sigma(b-\mu)}{b(\alpha+\mu)(\sigma+\mu)} \tag{7}$$

At equilibrium, the theoretical, mean field model demonstrates one of three infection states: endemic equilibrium, stable limit cycles, or no infection (*Figure 2*). Respectively, these states approximate the persistent infection, virus-induced epidemic extinction, and immune-mediated epidemic extinction phenotypes previously witnessed in tissue culture experiments (*Figure 1*). Theoretically, endemic equilibrium is maintained when new infections are generated at the same rate at which infections are lost, while limit cycles represent parameter space under which infectious and susceptible populations are locked in predictable oscillations. Endemic equilibria resulting from cellular regeneration (i.e. births) have been described *in vivo* for HIV (*Coffin, 1995*) and *in vitro* for herpesvirus plaque assays (*Howat et al., 2006*), but, because they so closely approach zero, true limit cycles likely only occur theoretically, instead yielding stochastic extinctions in empirical time series.

Bifurcation analysis of our mean field model revealed that regions of no infection (pathogen extinction) were bounded at lower threshold (Branch point) values for β, below which the pathogen was unable to invade. We found no upper threshold to invasion for β under any circumstances (i.e. β high enough to drive pathogen-induced extinction), but high β values resulted in Hopf bifurcations, which delineate regions of parameter space characterized by limit cycles. Since limit cycles so closely approach zero, high βs recovered in this range would likely produce virus-induced epidemic extinctions under experimental conditions. Under more robust representations of immunity, with higher values for either or both induced (ρ) and constitutive ($\varepsilon$) rates of antiviral acquisition, Hopf bifurcations occurred at increasingly higher values for β, meaning that persistent infections could establish at higher viral transmission rates (*Figure 2*). Consistent with our derivation for $R_0$, we found that the Branch point threshold for viral invasion was independent of changes to the induced immune parameter (ρ) but saturated at high values of $\varepsilon$ that characterize highly constitutive immunity (*Figure 3*).

We next fit our theoretical model by least squares to each cell line-virus combination, under absent, induced, and constitutive assumptions of immunity. In general, best fit models recapitulated expected outcomes based on the immune phenotype of the cell line in question, as described in the general literature (*Table 1*; *Supplementary file 4*). The absent immune model offered the most accurate approximation of IFN-deficient Vero cell time series, the induced immune model best

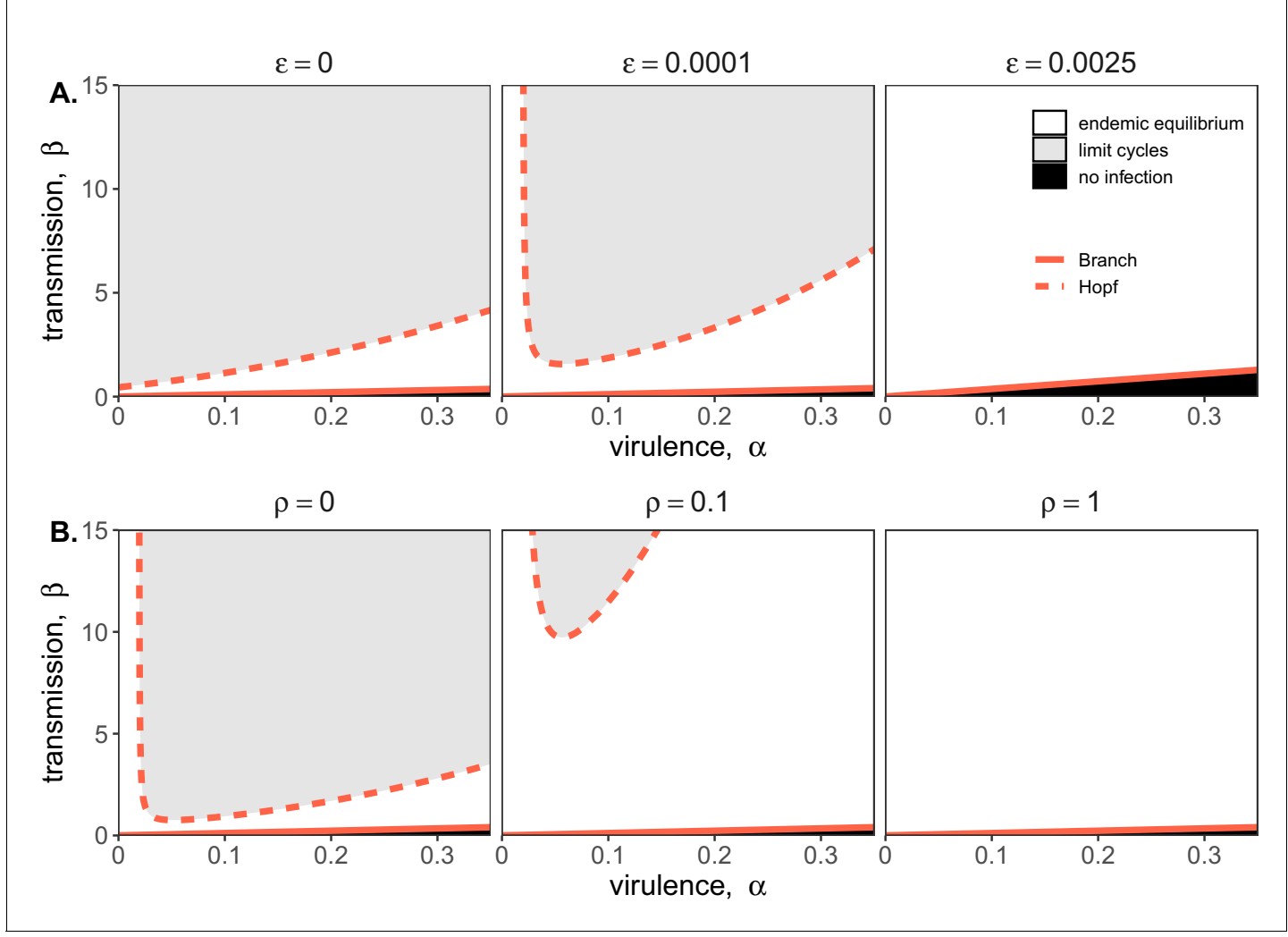

**Figure 2.** Two parameter bifurcations of the mean field model, showing variation in the transmission rate, β, against variation in the pathogen-induced mortality rate, α, under diverse immune assumptions. Panel (A) depicts dynamics under variably constitutive immunity, ranging from absent (left: $\varepsilon = 0$) to high (right: $\varepsilon = .0025$). In all panel (A) plots, the rate of induced immune antiviral acquisition (ρ) was fixed at 0.01. Panel (B) depicts dynamics under variably induced immunity, ranging from absent (left: ρ=0) to high (right: ρ=1). In all panel (B) plots, the rate of constitutive antiviral acquisition (ε) was fixed at 0.0001 Branch point curves are represented as solid lines and Hopf curves as dashed lines. White space indicates endemic equilibrium (persistence), gray space indicates limit cycles, and black space indicates no infection (extinction). Other parameter values for equilibrium analysis were fixed at: $b = .025$, $\mu = .001$, $\sigma = 1/6$, $c = 0$. Special points from bifurcations analyses are listed in *Supplementary file 3*.

recovered the RoNi/7.1 cell trials, and, in most cases, the constitutive immune model most closely recaptured infection dynamics across constitutively IFN-α-expressing PaKiT01 cell lines (*Figure 1*, *Figure 1—figure supplements 4–5*, *Supplementary file 4*). Ironically, the induced immune model offered a slightly better fit than the constitutive to rVSV-MARV infections on the PaKiT01 cell line (the one cell line-virus combination for which we know a constitutively antiviral cell-receptor incompatibility to be at play). Because constitutive immune assumptions can prohibit pathogen invasion ($R_0<1$), model fits to this time series under constitutive assumptions were handicapped by overestimations of $\varepsilon$, which prohibited pathogen invasion. Only by incorporating an exceedingly rapid rate of induced antiviral acquisition could the model guarantee that initial infection would be permitted and then rapidly controlled.

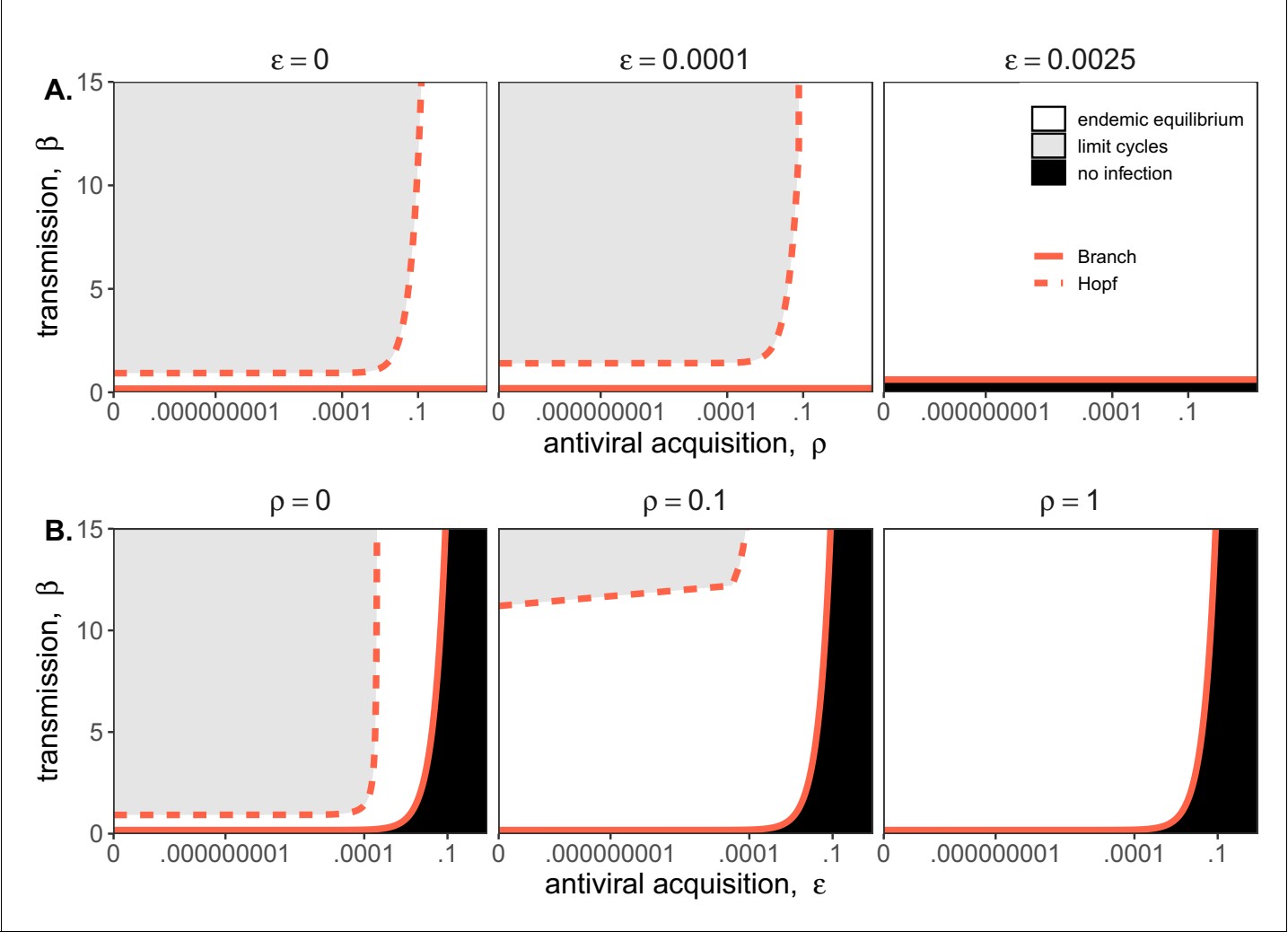

**Figure 3.** Two parameter bifurcations of the mean field model, showing variation in the transmission rate, β, against variation in: (A) the induced immunity rate of antiviral acquisition (ρ) and (B) the constitutive immunity rate of antiviral acquisition (ε). Panels show variation in the extent of immunity, from absent (left) to high (right). Branch point curves are represented as solid lines and Hopf curves as dashed lines. White space indicates endemic equilibrium (persistence), gray space indicates limit cycling, and black space indicates no infection (extinction). Other parameter values for equilibrium analysis were fixed at: b = .025, μ = .001, σ = 1/6, α = 1/6, c = 0. Special points from bifurcations analyses are listed in *Supplementary file 3*.

### Robust immunity is linked to rapid within-host virus transmission rates in fitted models

In fitting our theoretical model to in vitro data, we estimated the within-host virus transmission rate (β) and the rate(s) of cellular acquisition to antiviral status (ρ or ρ + ε) (*Table 1*; *Supplementary file 4*). Under absent immune assumptions, ρ and ε were fixed at 0 while β was estimated; under induced immune assumptions, ε was fixed at 0 while ρ and β were estimated; and under constitutive immune assumptions, all three parameters (ρ, ε, and β) were simultaneously estimated for each cell-virus combination. Best fit parameter estimates for MOI=0.001 data are visualized in conjunction with β – ρ and β – ε bifurcations in *Figure 4*; all general patterns were recapitulated at lower values for β on MOI=0.0001 trials (*Figure 4—figure supplement 1*).

As anticipated, the immune absent model (a simple target cell model) offered the best fit to IFN-deficient Vero cell infections (*Figure 4*; *Table 1*; *Supplementary file 4*). Among Vero cell trials, infections with rVSV-G produced the highest β estimates, followed by infections with rVSV-EBOV and rVSV-MARV. Best fit parameter estimates on Vero cell lines localized in the region of parameter

**Table 1.** Optimized parameters from best fit deterministic model and spatial approximation at MOI = 0.001

| Cell line | Virus | Immune assumption | AIC reduction from next-best model | Antiviral rate | $\varepsilon$ [lci – uci] * | $\rho$ [lci – uci] * | $\beta$ [lci – uci] * | Mean field $R_0$ | Spatial $\beta$ |
|---|---|---|---|---|---|---|---|---|---|
| Vero | rVSV-G | Absent | 2 | 0 | 0 [0–0] | 0 [0–0] | 2.44 [1.52–3.36] | 8.73 | 24.418 |
| | rVSV-EBOV | Absent | 2 | 0 | 0 [0–0] | 0 [0–0] | 1.5 [1.06–1.94] | 5.42 | 14.996 |
| | rVSV-MARV | Absent | 2 | 0 | 0 [0–0] | 0 [0–0] | 0.975 [0.558–1.39] | 3.45 | 9.752 |
| RoNi/7.1 | rVSV-G | Induced | 2 | $7.03 \times 10^{-5}$ | 0 [0–0] | 0.089 [0–0.432] | 2.47 [1.49–3.45] | 10.91 | 24.705 |
| | rVSV-EBOV | Induced | 2.01 | $2.87 \times 10^{-5}$ | 0 [0–0] | 0.0363 [0–0.343] | 0.685 [0.451–0.919] | 3.04 | 6.849 |
| | rVSV-MARV | Induced | 2 | $1.40 \times 10^{-5}$ | 0 [0–0] | 0.0177 [0–0.257] | 1.23 [0.917–1.55] | 5.48 | 12.324 |
| PaKiT01 | rVSV-G | Constitutive | 29.9 | .00209 | 0.00602 [0–0.019] | $8.26 \times 10^{-8}$ [0–$4.75 \times 10^{-7}$] | 3.45 [1.07–5.84] | 6.20 | 34.516 |
| | rVSV-EBOV | Constitutive | 27.9 | .00499 | 0.0478 [0–0.0958] | $4.46 \times 10^{-8}$ [0–$4.37 \times 10^{-7}$] | 34.5 [28.7–40.2] | 18.82 | 344.821 |
| | rVSV-MARV | Induced | 2 | .00687 | 0 [0–0] | 13.1 [0–37.9] | 3.25 [0–41.3] | 8.83 | 32.452 |

Improvement in AIC from next best model for same cell line-virus-MOI combination. All δ-AIC are reported in **Supplementary file 4**.

*lci = lower and uci = upper 95% confidence interval. No confidence interval is shown for spatial β which was fixed at 10 times the estimated mean for the mean field model fits when paired with equivalent values of $\varepsilon$ and ρ.

All other parameters were fixed at: b = 0.025 (mean field), 0.15 (spatial); $\alpha$ = 1/6; c = 0; $\mu$ = 1/121 (Vero), 1/191 (RoNi/7.1), and 1/84 (PaKiT01).

space corresponding to theoretical limit cycles, consistent with observed virus-induced epidemic extinctions in stochastic tissue cultures.

In contrast to Vero cells, the induced immunity model offered the best fit to all RoNi/7.1 data, consistent with reported patterns in the literature and our own validation by qPCR (*Table 1*; *Figure 1—figure supplement 6*; *Arnold et al., 2018*; *Kuzmin et al., 2017*; *Biesold et al., 2011*; *Pavlovich et al., 2018*). As in Vero cell trials, we estimated highest β values for rVSV-G infections on RoNi/7.1 cell lines but here recovered higher β estimates for rVSV-MARV than for rVSV-EBOV. This reversal was balanced by a higher estimated rate of acquisition to antiviral status (ρ) for rVSV-EBOV versus rVSV-MARV. In general, we observed that more rapid rates of antiviral acquisition (either induced, ρ, constitutive, $\varepsilon$, or both) correlated with higher transmission rates (β). When offset by ρ, β values estimated for RoNi/7.1 infections maintained the same amplitude as those estimated for immune-absent Vero cell lines but caused gentler epidemics and reduced cellular mortality (*Figure 1*). RoNi/7.1 parameter estimates localized in the region corresponding to endemic equilibrium for the deterministic, theoretical model (*Figure 4*), yielding less acute epidemics which nonetheless went extinct in stochastic experiments.

Finally, rVSV-G and rVSV-EBOV trials on PaKiT01 cells were best fit by models assuming constitutive immunity, while rVSV-MARV infections on PaKiT01 were matched equivalently by models assuming either induced or constitutive immunity—with induced models favored over constitutive in AIC comparisons because one fewer parameter was estimated (*Figure 1—figure supplements 4–5*; *Supplementary file 4*). For all virus infections, PaKiT01 cell lines yielded β estimates a full order of magnitude higher than Vero or RoNi/7.1 cells, with each β balanced by an immune response (either ρ, or ρ combined with $\varepsilon$) also an order of magnitude higher than that recovered for the other cell lines (*Figure 4*; *Table 1*). As in RoNi/7.1 cells, PaKiT01 parameter fits localized in the region corresponding to endemic equilibrium for the deterministic theoretical model. Because constitutive immune processes can actually prohibit initial pathogen invasion, constitutive immune fits to rVSV-MARV infections on PaKiT01 cell lines consistently localized at or below the Branch point threshold for virus invasion ($R_0$ = 1). During model fitting for optimization of $\varepsilon$, any parameter tests of $\varepsilon$ values producing $R_0 < 1$ resulted in no infection and, consequently, produced an exceedingly poor fit to infectious time series data. In all model fits assuming constitutive immunity, across all cell lines,

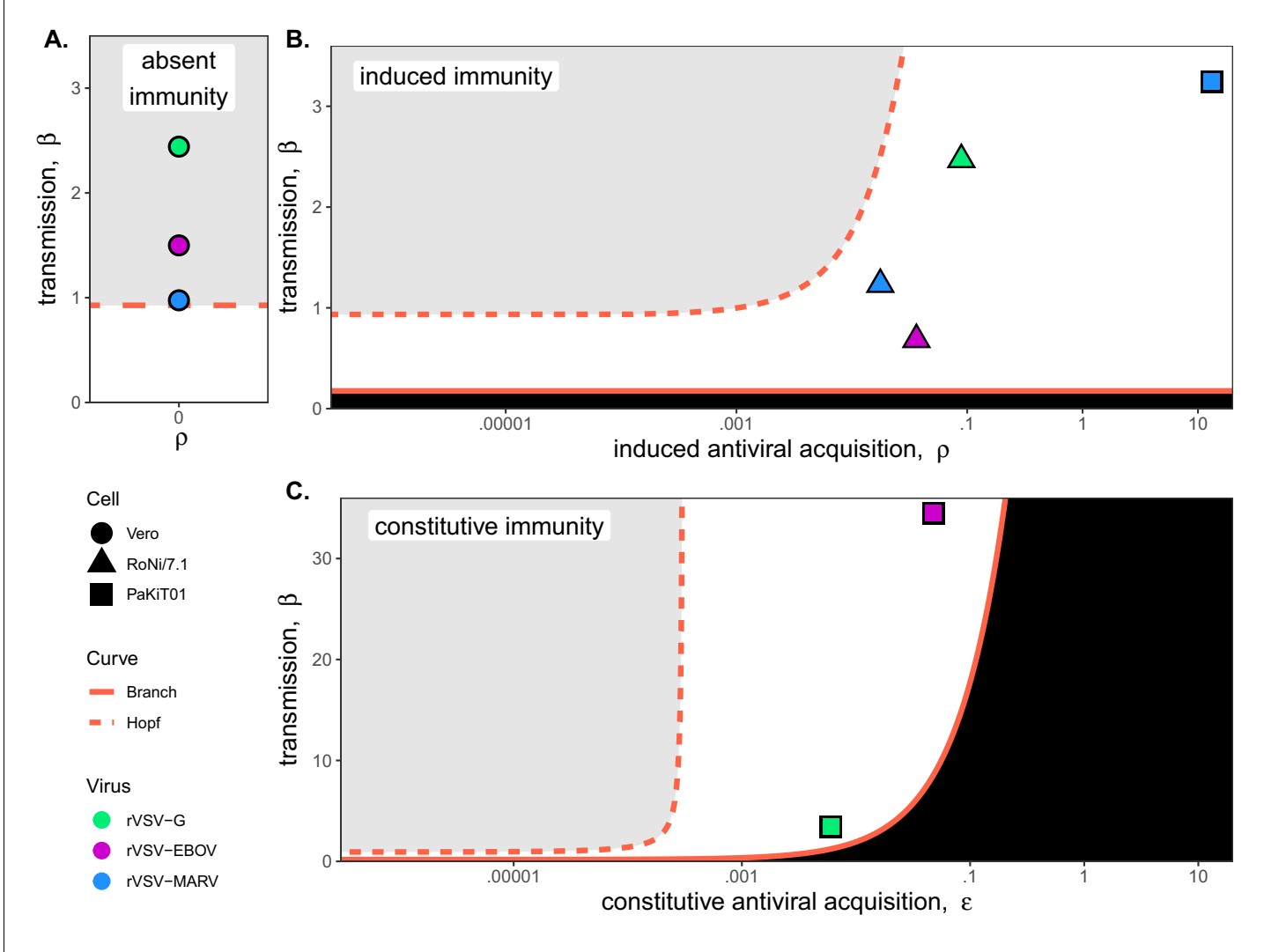

**Figure 4.** Best fit parameter estimates for β and ρ or ε from mean-field model fits to MOI=0.001 time series data, atop (A,B) β – ρ and (C) β – ε bifurcation. Fits and bifurcations are grouped by immune phenotype: (A) absent; (B) induced; (C) constitutive immunity, with cell lines differentiated by shape (Vero=circles; RoNi/7.1 = triangles; PaKiT01=squares) and viral infections by color (rVSV-G = green, rVSV-EBOV = magenta, rVSV-MARV = blue). Note that y-axis values are ten-fold higher in panel (C). Branch point curves (solid lines) and Hopf curves (dashed lines) are reproduced from *Figure 3*. White space indicates endemic equilibrium (pathogen persistence), gray space indicates limit cycling (virus-induced epidemic extinction), and black space indicates no infection (immune-mediated pathogen extinction). In panel (A) and (B), ε is fixed at 0; in panel (C), ρ is fixed at 5x10$^{-8}$ for bifurcation curves and estimated at 4x10$^{-8}$ and 8x10$^{-8}$ for rVSV-EBOV and rVSV-G parameter points, respectively. Other parameter values were fixed at: $b$ = .025, μ = 0.001, σ = 1/6, α = 1/6, and c = 0 across all panels. Raw fitted values and corresponding 95% confidence intervals for β, ρ, and ε, background parameter values, and AIC recovered from model fit, are reported in *Supplementary file 4*. Parameter fits at MOI=0.0001 are visualized in *Figure 4— figure supplement 1*.

The online version of this article includes the following figure supplement(s) for figure 4:

**Figure supplement 1.** Best fit parameter estimates for β and ρ or ε from mean-field model fits to MOI=0.0001 time series data, atop (A,B) β – ρ and (C) β – ε bifurcation.

parameter estimates for ρ and ε traded off, with one parameter optimized at values approximating zero, such that the immune response was modeled as almost entirely induced or entirely constitutive (*Table 1*; *Supplementary file 4*). For RoNi/7.1 cells, even when constitutive immunity was allowed, the immune response was estimated as almost entirely induced, while for rVSV-G and rVSV-EBOV fits on PaKiT01 cells, the immune response optimized as almost entirely constitutive. For rVSV-MARV on PaKiT01 cells, however, estimation of ρ was high under all assumptions, such that any additional

antiviral contributions from $\varepsilon$ prohibited virus from invading at all. The induced immune model thus produced a more parsimonious recapitulation of these data because virus invasion was always permitted, then rapidly controlled.

## Antiviral cells safeguard live cells against rapid cell mortality to elongate epidemic duration *in vitro*

In order to compare the relative contributions of each cell line's disparate immune processes to epidemic dynamics, we next used our mean field parameter estimates to calculate the initial 'antiviral rate'—the initial accumulation rate of antiviral cells upon virus invasion for each cell-virus-MOI combination—based on the following equation:

$$Antiviral\,Rate = \rho P_E P_s - \epsilon P_s \tag{8}$$

where $P_E$ was calculated from the initial infectious dose (MOI) of each infection experiment and $P_S$ was estimated at disease-free equilibrium:

$$P_E = 1 - e^{-MOI} \tag{9}$$

$$P_S = \frac{(b-\mu)(c+\mu)}{b(c+\mu+\varepsilon)} \tag{10}$$

Because $\rho$ and $\varepsilon$ both contribute to this initial antiviral rate, induced and constitutive immune assumptions are capable of yielding equally rapid rates, depending on parameter fits. Indeed, under fully induced immune assumptions, the induced antiviral acquisition rate ($\rho$) estimated for rVSV-MARV infection on PaKiT01 cells was so high that the initial antiviral rate exceeded even that estimated under constitutive assumptions for this cell-virus combination (*Supplementary file 4*). In reality, we know that NPC1 receptor incompatibilities make PaKiT01 cell lines constitutively refractory to rVSV-MARV infection (Ng and Chandrab, 2018, Unpublished results) and that PaKiT01 cells also constitutively express the antiviral cytokine, IFN-α. Model fitting results suggest that this constitutive expression of IFN-α may act more as a rapidly inducible immune response following virus invasion than as a constitutive secretion of functional IFN-α protein. Nonetheless, as hypothesized, PaKiT01 cell lines were by far the most antiviral of any in our study—with initial antiviral rates estimated several orders of magnitude higher than any others in our study, under either induced or constitutive assumptions (*Table 1*; *Supplementary file 4*). RoNi/7.1 cells displayed the second-most-pronounced signature of immunity, followed by Vero cells, for which the initial antiviral rate was essentially zero even under forced assumptions of induced or constitutive immunity (*Table 1*; *Supplementary file 4*).

Using fitted parameters for $\beta$ and $\varepsilon$, we additionally calculated $R_0$, the basic reproduction number for the virus, for each cell line-virus-MOI combination (*Table 1*; *Supplementary file 4*). We found that $R_0$ was essentially unchanged across differing immune assumptions for RoNi/7.1 and Vero cells, for which the initial antiviral rate was low. In the case of PaKiT01 cells, a high initial antiviral rate under either induced or constitutive immunity resulted in a correspondingly high estimation of $\beta$ (and, consequently, $R_0$) which still produced the same epidemic curve that resulted from the much lower estimates for $\beta$ and $R_0$ paired with absent immunity. These findings suggest that antiviral immune responses protect host tissues against virus-induced cell mortality and may facilitate the establishment of more rapid within-host transmission rates.

Total monolayer destruction occurred in all cell-virus combinations excepting rVSV-EBOV infections on RoNi/7.1 cells and rVSV-EBOV and rVSV-MARV infections on PaKiT01 cells. Monolayer destruction corresponded to susceptible cell depletion and epidemic turnover where R-effective (the product of $R_0$ and the proportion susceptible) was reduced below one (*Figure 5*). For rVSV-EBOV infections on RoNi/7.1, induced antiviral cells safeguarded remnant live cells, which birthed new susceptible cells late in the time series. In rVSV-EBOV and rVSV-MARV infections on PaKiT01 cells, this antiviral protection halted the epidemic (*Figure 5*; R-effective <1) before susceptibles fully declined. In the case of rVSV-EBOV on PaKiT01, the birth of new susceptibles from remnant live cells protected by antiviral status maintained late-stage transmission to facilitate long-term epidemic persistence. Importantly, under fixed parameter values for the infection incubation rate ($\sigma$) and infection-induced mortality rate ($\alpha$), models were unable to reproduce the longer-term infectious time series

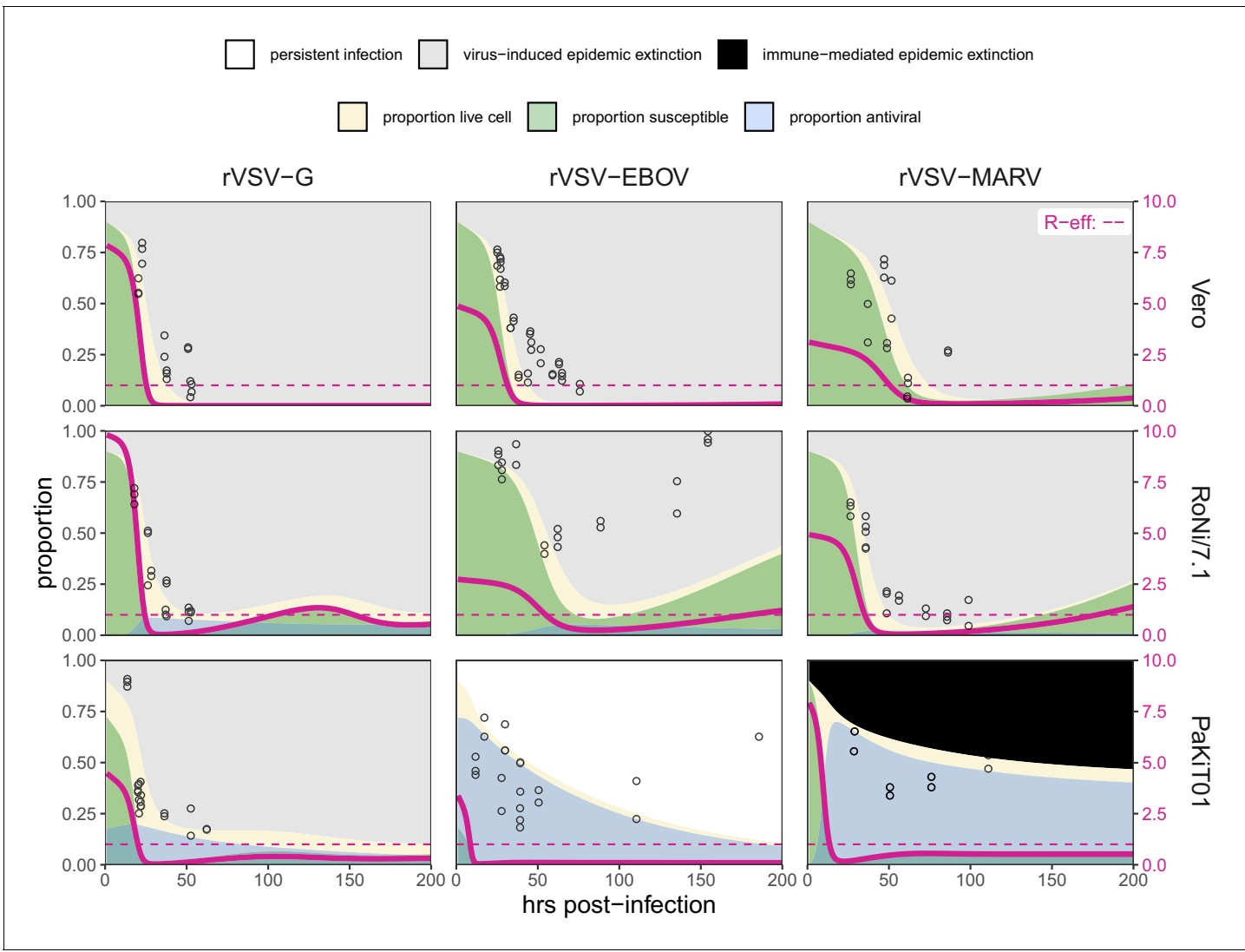

**Figure 5.** Fitted time series of susceptible (green shading) and antiviral (blue shading) cell proportions from the mean field model for rVSV-G, rVSV-EBOV, and rVSV-MARV infections (columns) on Vero, RoNi/7.1, and PaKiT01 cell lines (rows) at MOI = 0.001. Results are shown for the best fit immune absent model on Vero cells, induced immunity model on RoNi/7.1 cells and constitutive (rVSV-G and rVSV-EBOV) and induced (rVSV-MARV) immune models on PaKiT01 cells. Combined live, uninfectious cell populations (S + A + E) are shown in tan shading, with raw live, uninfectious cell data from Hoechst stains visualized as open circles. The right-hand y-axis corresponds to R-effective (pink solid line) across each time series; R-effective = 1 is a pink dashed, horizontal line. Panel background corresponds to empirical outcome of the average stochastic cell culture trial (persistent infection = white; virus-induced epidemic extinction = gray; immune-mediated epidemic extinction = black). Parameter values are listed in *Supplementary file 4* and results for absent/induced/constitutive fitted models across all cell lines in *Figure 5—figure supplement 1* (MOI = 0.001) and *Figure 5—figure supplement 2* (MOI = 0.0001).

The online version of this article includes the following figure supplement(s) for figure 5:

**Figure supplement 1.** Figure replicates *Figure 5* (main text) but includes all output across mean field model fits assuming (A) absent immunity, (B) induced immunity, and (C) constitutive immunity.

**Figure supplement 2.** Figure replicates *Figure 5—figure supplement 1* exactly but shows model fits and data for all cell-virus combinations at MOI = 0.0001.

**Figure supplement 3.** Spatial model state variable outputs, fit to MOI = 0.001 data only, for all 27 unique cell line - virus - immune assumption combinations: (A) absent immunity, (B) induced immunity, and (C) constitutive immunity.

captured in data from rVSV-EBOV infections on PaKiT01 cell lines without incorporation of cell births, an assumption adopted in previous modeling representations of IFN-mediated viral dynamics in tissue culture (*Howat et al., 2006*). In our experiments, we observed that cellular reproduction took place as plaque assays achieved confluency.

Finally, because the protective effect of antiviral cells is more clearly observable spatially, we confirmed our results by simulating fitted time series in a spatially-explicit, stochastic reconstruction of our mean field model. In spatial simulations, rates of antiviral acquisition were fixed at fitted values for ρ and $\varepsilon$ derived from mean field estimates, while transmission rates (β) were fixed at values ten times greater than those estimated under mean field conditions, accounting for the intensification of parameter thresholds permitting pathogen invasion in local spatial interactions (see Materials and methods; *Videos 1–3*; *Figure 5—figure supplement 3*; *Supplementary file 5*; *Webb et al., 2007*). In immune capable time series, spatial antiviral cells acted as 'refugia' which protected live cells from infection as each initial epidemic wave 'washed' across a cell monolayer. Eventual birth of new susceptibles from these living refugia allowed for sustained epidemic transmission in cases where some infectious cells persisted at later timepoints in simulation (*Videos 1–3*; *Figure 5—figure supplement 3*).

## Discussion

Bats are reservoirs for several important emerging zoonoses but appear not to experience disease from otherwise virulent viral pathogens. Though the molecular biological literature has made great progress in elucidating the mechanisms by which bats tolerate viral infections (*Zhou et al., 2016*; *Ahn et al., 2019*; *Xie et al., 2018*; *Pavlovich et al., 2018*; *Zhang et al., 2013*), the impact of unique bat immunity on virus dynamics within-host has not been well-elucidated. We used an innovative combination of *in vitro* experimentation and within-host modeling to explore the impact of unique bat immunity on virus dynamics. Critically, we found that bat cell lines demonstrated a signature of enhanced interferon-mediated immune response, of either constitutive or induced form, which allowed for establishment of rapid within-host, cell-to-cell virus transmission rates (β). These results were supported by both data-independent bifurcation analysis of our mean field theoretical model, as well as fitting of this model to viral infection time series established in bat cell culture. Additionally, we demonstrated that the antiviral state induced by the interferon pathway protects live cells from mortality in tissue culture, resulting in *in vitro* epidemics of extended duration that enhance the probability of establishing a long-term persistent infection. Our findings suggest that viruses evolved in bat reservoirs possessing enhanced IFN capabilities could achieve more rapid within-host transmission rates without causing pathology to their hosts. Such rapidly-reproducing viruses would likely generate extreme virulence upon spillover to hosts lacking similar immune capacities to bats.

To achieve these results, we first developed a novel, within-host, theoretical model elucidating the effects of unique bat immunity, then undertook bifurcation analysis of the model's equilibrium properties under immune absent, induced, and constitutive assumptions. We considered a cell line to be constitutively immune if possessing any number of antiviral cells at disease-free equilibrium but allowed the extent of constitutive immunity to vary across the parameter range for $\varepsilon$, the constitutive rate of antiviral acquisition. In deriving the equation for $R_0$, the basic reproduction number, which defines threshold conditions for virus invasion of a tissue ($R_0 > 1$), we demonstrated how the invasion threshold is elevated at high values of constitutive antiviral acquisition, $\varepsilon$. Constitutive immune processes can thus prohibit pathogen invasion, while induced responses, by definition, can only control infections *post-hoc*. Once thresholds for pathogen invasion have been met, assumptions of constitutive immunity will limit the cellular mortality (virulence) incurred at high transmission rates. Regardless of mechanism (induced or constitutive), interferon-stimulated antiviral cells appear to play a key role in maintaining longer term or persistent infections by safeguarding susceptible cells from rapid infection and concomitant cell death.

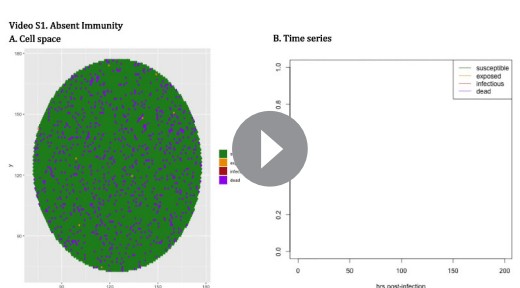

**Video 1.** Two hundred hour time series of spatial stochastic model for rVSV-EBOV infection on 10,000 cell grid for PaKiT01, assuming conditions of *absent immunity*: (A) spatial spread of infection, (B) time series of state variables. Parameter values are listed in *Supplementary file 4*.

https://elifesciences.org/articles/48401#video1

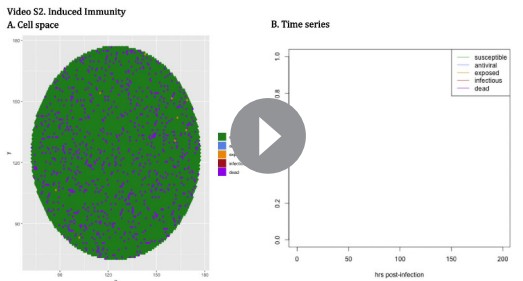

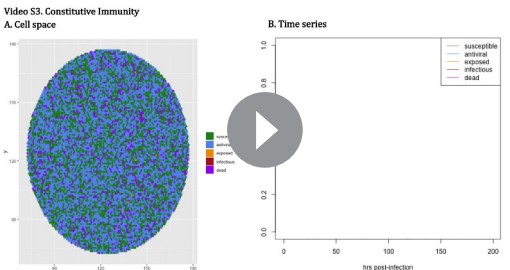

**Video 2.** Two hundred hour time series of spatial stochastic model for rVSV-EBOV infection on 10,000 cell grid for PaKiT01, assuming conditions of *induced immunity*: (A) spatial spread of infection, (B) time series of state variables. Parameter values are listed in *Supplementary file 4*.
https://elifesciences.org/articles/48401#video2

**Video 3.** Two hundred hour time series of spatial stochastic model for rVSV-EBOV infection on 10,000 cell grid for PaKiT01, assuming conditions of *constitutive immunity*: (A) spatial spread of infection, (B) time series of state variables. Parameter values are listed in *Supplementary file 4*.
https://elifesciences.org/articles/48401#video3

Fitting of our model to *in vitro* data supported expected immune phenotypes for different bat cell lines as described in the literature. Simple target cell models that ignore the effects of immunity best recapitulated infectious time series derived from IFN-deficient Vero cells, while models assuming induced immune processes most accurately reproduced trials derived from RoNi/7.1 (*Rousettus aegyptiacus*) cells, which possess a standard virus-induced IFN-response. In most cases, models assuming constitutive immune processes best recreated virus epidemics produced on PaKiT01 (*Pteropus alecto*) cells, which are known to constitutively express the antiviral cytokine, IFN-α (*Zhou et al., 2016*). Model support for induced immune assumptions in fits to rVSV-MARV infections on PaKiT01 cells suggests that the constitutive IFN-α expression characteristic of *P. alecto* cells may represent more of a constitutive immune priming process than a perpetual, functional, antiviral defense. Results from mean field model fitting were additionally confirmed in spatially explicit stochastic simulations of each time series.

As previously demonstrated in within-host models for HIV (*Coffin, 1995*; *Perelson et al., 1996*; *Nowak et al., 1995*; *Bonhoeffer et al., 1997*; *Ho et al., 1995*), assumptions of simple target-cell depletion can often provide satisfactory approximations of viral dynamics, especially those reproduced in simple *in vitro* systems. Critically, our model fitting emphasizes the need for incorporation of top-down effects of immune control in order to accurately reproduce infectious time series derived from bat cell tissue cultures, especially those resulting from the robustly antiviral PaKiT01 *P. alecto* cell line. These findings indicate that enhanced IFN-mediated immune pathways in bat reservoirs may promote elevated within-host virus replication rates prior to cross-species emergence. We nonetheless acknowledge the limitations imposed by *in vitro* experiments in tissue culture, especially involving recombinant viruses and immortalized cell lines. Future work should extend these cell culture studies to include measurements of multiple state variables (i.e. antiviral cells) to enhance epidemiological inference.

The continued recurrence of Ebola epidemics across central Africa highlights the importance of understanding bats' roles as reservoirs for virulent zoonotic disease. The past decade has born witness to emerging consensus regarding the unique pathways by which bats resist and tolerate highly virulent infections (*Brook and Dobson, 2015*; *Xie et al., 2018*; *Zhang et al., 2013*; *Ahn et al., 2019*; *Zhou et al., 2016*; *Ng et al., 2015*; *Pavlovich et al., 2018*). Nonetheless, an understanding of the mechanisms by which bats support endemic pathogens at the population level, or promote the evolution of virulent pathogens at the individual level, remains elusive. Endemic maintenance of infection is a defining characteristic of a pathogen reservoir (*Haydon et al., 2002*), and bats appear to merit such a title, supporting long-term persistence of highly transmissible viral infections in isolated island populations well below expected critical community sizes (*Peel et al., 2012*). Researchers debate the relative influence of population-level and within-host mechanisms which might explain these trends (*Plowright et al., 2016*), but increasingly, field data are difficult to reconcile without acknowledgment of a role for persistent infections (*Peel et al., 2018*; *Brook et al., 2019*).

We present general methods to study cross-scale viral dynamics, which suggest that within-host persistence is supported by robust antiviral responses characteristic of bat immune processes. Viruses which evolve rapid replication rates under these robust antiviral defenses may pose the greatest hazard for cross-species pathogen emergence into spillover hosts with immune systems that differ from those unique to bats.

# Materials and methods

## Key resources table

| Reagent type (species) or resource | Designation | Source or reference | Identifiers | Additional information |
|---|---|---|---|---|
| Cell line (Vero) | Kidney (normal, epithelial, adult) | ATCC | CCL-81 | |
| Cell line (*Rousettus aegyptiacus*) | Kidney (normal, epithelial, adult) | (*Biesold et al., 2011*; *Kühl et al., 2011*) | RoNi/7.1 | |
| Cell line (*Pteropus alecto*) | Kidney (normal, epithelial, adult) | (*Crameri et al., 2009*) | PaKiT01 | |
| Virus strain | Replication competent, recombinant vesicular stomatitis Indiana virus expressing eGFP | (*Miller et al., 2012*; *Wong et al., 2010*) | rVSV-G | |
| Virus strain | Replication competent, recombinant vesicular stomatitis Indiana virus expressing eGFP and EBOV GP in place of VSV G | (*Miller et al., 2012*; *Wong et al., 2010*) | rVSV-EBOV | |
| Virus strain | Replication competent, recombinant vesicular stomatitis Indiana virus expressing eGFP and MARV GP in place of VSV G | (*Miller et al., 2012*; *Wong et al., 2010*) | rVSV-MARV | |
| Reagent | Hoechst 33342 Fluorescent Stain | ThermoFisher | cat #: 62249 | |
| Reagent | L-Glutamine Solution | ThermoFisher | cat #: 25030081 | |
| Reagent | Gibco HEPES | ThermoFisher | cat #: 15630080 | |
| Reagent | iTaq Universal SYBR Green Supermix | BioRad | cat #: 1725120 | |
| Commercial assay or kit | Quick RNA Mini Prep Kit | Zymo | cat #: R1054 | |
| Commercial assay or kit | Invitrogen Superscript III cDNA Synthesis Kit | ThermoFisher | cat #: 18080051 | |
| Software | MatCont (version 2.2) | (*Dhooge et al., 2008*) | MatCont | |
| R | R version 3.6.0 | (*R Development Core Team, 2019*) | R | |

*Note that primers for *R. aegyptiacus* and *P. alecto* β-Actin, IFN-α, and IFN-β genes are listed in the *Supplementary file 6*.

## Cell culture experiments
### Cells
All experiments were carried out on three immortalized mammalian kidney cell lines: Vero (African green monkey), RoNi/7.1 (*Rousettus aegyptiacus*) (*Kühl et al., 2011*; *Biesold et al., 2011*) and PaKiT01 (*Pteropus alecto*) (*Crameri et al., 2009*). The species identifications of all bat cell lines was confirmed morphologically and genetically in the publications in which they were originally described (*Kühl et al., 2011*; *Biesold et al., 2011*; *Crameri et al., 2009*). Vero cells were obtained from ATCC.

Monolayers of each cell line were grown to 90% confluency (~$9 \times 10^5$ cells) in 6-well plates. Cells were maintained in a humidified 37 °C, 5% $CO_2$ incubator and cultured in Dulbecco's modified Eagle medium (DMEM) (Life Technologies, Grand Island, NY), supplemented with 2% fetal bovine serum (FBS) (Gemini Bio Products, West Sacramento, CA), and 1% penicillin-streptomycin (Life Technologies). Cells were tested monthly for mycoplasma contamination while experiments were taking place; all cells assayed negative for contamination at every testing.

Previous work has demonstrated that all cell lines used are capable of mounting a type I IFN response upon viral challenge, with the exception of Vero cells, which possess an IFN-β deficiency (*Desmyter et al., 1968*; *Rhim et al., 1969*; *Emeny and Morgan, 1979*). RoNi/7.1 cells have been shown to mount idiosyncratic induced IFN defenses upon viral infection (*Pavlovich et al., 2018*; *Kuzmin et al., 2017*; *Arnold et al., 2018*; *Kühl et al., 2011*; *Biesold et al., 2011*), while PaKiT01 cells are known to constitutively express the antiviral cytokine, IFN-α (*Zhou et al., 2016*). This work is the first documentation of IFN signaling induced upon challenge with the particular recombinant VSVs outlined below. We verified known antiviral immune phenotypes via qPCR. Results were consistent with the literature, indicating a less pronounced role for interferon defense against viral infection in RoNi/7.1 versus PaKiT01 cells.

## Viruses

Replication-capable recombinant vesicular stomatitis Indiana viruses, expressing filovirus glycoproteins in place of wild type G (rVSV-G, rVSV-EBOV, and rVSV-MARV) have been previously described (*Wong et al., 2010*; *Miller et al., 2012*). Viruses were selected to represent a broad range of anticipated antiviral responses from host cells, based on a range of past evolutionary histories between the virus glycoprotein mediating cell entry and the host cell's entry receptor. These interactions ranged from the total absence of evolutionary history in the case of rVSV-G infections on all cell lines to a known receptor-level cell entry incompatibility in the case of rVSV-MARV infections on PaKiT01 cell lines.

To measure infectivities of rVSVs on each of the cell lines outlined above, so as to calculate the correct viral dose for each MOI, $NH_4Cl$ (20 mM) was added to infected cell cultures at 1–2 hr post-infection to block viral spread, and individual eGFP-positive cells were manually counted at 12–14 hr post-infection.

## Innate immune phenotypes via qPCR of IFN genes

Previously published work indicates that immortalized kidney cell lines of *Rousettus aegyptiacus* (RoNi/7.1) and *Pteropus alecto* (PaKiT01) exhibit different innate antiviral immune phenotypes through, respectively, induced (*Biesold et al., 2011*; *Pavlovich et al., 2018*; *Kühl et al., 2011*; *Arnold et al., 2018*) and constitutive (*Zhou et al., 2016*) expression of type I interferon genes. We verified these published phenotypes on our own cell lines infected with rVSV-G, rVSV-EBOV, and rVSV-MARV via qPCR of IFN-α and IFN-β genes across a longitudinal time series of infection.

Specifically, we carried out multiple time series of infection of each cell line with each of the viruses described above, under mock infection conditions and at MOIs of 0.0001 and 0.001—with the exception of rVSV-MARV on PaKiT01 cell lines, for which infection was only performed at MOI = 0.0001 due to limited viral stocks and the extremely low infectivity of this virus on this cell line (thus requiring high viral loads for initial infection). All experiments were run in duplicate on 6-well plates, such that a typical plate for any of the three viruses had two control (mock) wells, two MOI = 0.0001 wells and two MOI = 0.001 wells, excepting PaKiT01 plates, which had two control and four MOI = 0.0001 wells at a given time. We justify this PaKiT01 exemption through the expectation that IFN-α expression is constitutive for these cells, and by the assumption that any expression exhibited at the lower MOI should also be present at the higher MOI.

For these gene expression time series, four 6-well plates for each cell line–virus combination were incubated with virus for one hour at 37 °C. Following incubation, virus was aspirated off, and cell monolayers were washed in PBS, then covered with an agar plaque assay overlay to mimic conditions under which infection trials were run. Plates were then harvested sequentially at timepoints of roughly 5, 10, 15, and 20 hr post-infection (exact timing varied as multiple trials were running simultaneously). Upon harvest of each plate, agar overlay was removed, and virus was lysed and RNA extracted from cells using the Zymo Quick RNA Mini Prep kit, according to the manufacturer's

instructions and including the step for cellular DNA digestion. Post-extraction, RNA quality was verified via nanodrop, and RNA was converted to cDNA using the Invitrogen Superscript III cDNA synthesis kit, according to the manufacturer's instructions. cDNA was then stored at 4 °C and as a frozen stock at −20 °C to await qPCR.

We undertook qPCR of cDNA to assess expression of the type I interferon genes, IFN-α and IFN-β, and the housekeeping gene, β-Actin, using primers previously reported in the literature (*Supplementary file 6*). For qPCR, 2 µl of each cDNA sample was incubated with 7 µl of deionized water, 1 µl of 5 UM forward/reverse primer mix and 10 µl of iTaq Universal SYBR Green, then cycled on a QuantStudio3 Real-Time PCR machine under the following conditions: initial denaturation at 94 °C for 2 min followed by 40 cycles of: denaturation at 95 °C (5 s), annealing at 58 °C (15 s), and extension at 72 °C (10 s).

We report simple δ-Ct values for each run, with raw Ct of the target gene of interest (IFN-α or IFN-β) subtracted from raw Ct of the β-Actin housekeeping gene in *Figure 1—figure supplement 6*. Calculation of fold change upon viral infection in comparison to mock using the δ-δ-Ct method (*Livak and Schmittgen, 2001*) was inappropriate in this case, as we wished to demonstrate constitutive expression of IFN-α in PaKiT01 cells, whereby data from mock cells was identical to that produced from infected cells.

## Plaque assays and time series imaging

After being grown to ~90% confluency, cells were incubated with pelleted rVSVs expressing eGFP (rVSV-G, rVSV-EBOV, rVSV-MARV). Cell lines were challenged with both a low (0.0001) and high (0.001) multiplicity of infection (MOI) for each virus. In a cell monolayer infected at a given MOI (m), the proportion of cells (P), infected by k viral particles can be described by the Poisson distribution: $P(k) = \frac{e^{-m}m^k}{k!}$, such that the number of initially infected cells in an experiment equals: $1 - e^{-m}$. We assumed that a ~90% confluent culture at each trial's origin was comprised of ~9x10$^5$ cells and conducted all experiments at MOIs of 0.0001 and 0.001, meaning that we began each trial by introducing virus to, respectively, ~81 or 810 cells, representing the state variable 'E' in our theoretical model. Low MOIs were selected to best approximate the dynamics of mean field infection and limit artifacts of spatial structuring, such as premature epidemic extinction when growing plaques collide with plate walls in cell culture.

Six-well plates were prepared with each infection in duplicate or triplicate, such that a control well (no virus) and 2–3 wells each at MOI 0.001 and 0.0001 were incubated simultaneously on the same plate. In total, we ran between 18 and 39 trials at each cell-virus-MOI combination, excepting r-VSV-MARV infections on PaKiT01 cells at MOI = 0.001, for which we ran only eight trials due to the low infectivity of this virus on this cell line, which required high viral loads for initial infection. Cells were incubated with virus for one hour at 37 °C. Following incubation, virus was aspirated off, and cell monolayers were washed in PBS, then covered with a molten viscous overlay (50% 2X MEM/L-glutamine; 5% FBS; 3% HEPES; 42% agarose), cooled for 20 min, and re-incubated in their original humidified 37 °C, 5% CO$_2$ environment.

After application of the overlay, plates were monitored periodically using an inverted fluorescence microscope until the first signs of GFP expression were witnessed (~6–9.5 hr post-infection, depending on the cell line and virus under investigation). From that time forward, a square subset of the center of each well (comprised of either 64- or 36-subframes and corresponding to roughly 60% and 40% of the entire well space) was imaged periodically, using a CellInsight CX5 High Content Screening (HCS) Platform with a 4X air objective (ThermoFisher, Inc, Waltham, MA). Microscope settings were held standard across all trials, with exposure time fixed at 0.0006 s for each image. One color channel was imaged, such that images produced show GFP-expressing cells in white and non-GFP-expressing cells in black (*Figure 1—figure supplement 1*).

Wells were photographed in rotation, as frequently as possible, from the onset of GFP expression until the time that the majority of cells in the well were surmised to be dead, GFP expression could no longer be detected, or early termination was desired to permit Hoechst staining.

In the case of PaKiT01 cells infected with rVSV-EBOV, where an apparently persistent infection established, the assay was terminated after 200+ hours (8+ days) of continuous observation. Upon termination of all trials, cells were fixed in formaldehyde (4% for 15 min), incubated with Hoechst stain (0.0005% for 15 min) (ThermoFisher, Inc, Waltham, MA), then imaged at 4X on the CellInsight

CX5 High Content Screening (HCS) Platform. The machine was allowed to find optimal focus for each Hoechst stain image. One color channel was permitted such that images produced showed live nuclei in white and dead cells in black.

## Hoechst staining

Hoechst stain colors cellular DNA, and viral infection is thought to interfere with the clarity of the stain (*Dembowski and DeLuca, 2015*). As such, infection termination, cell fixation, and Hoechst staining enables generation of a rough time series of uninfectious live cells (i.e. susceptible + antiviral cells) to complement the images which produced time series of proportions infectious. Due to uncertainty over the exact epidemic state of Hoechst-stained cells (i.e. exposed but not yet infectious cells may still stain), we elected to fit our models only to the infectious time series derived from GFP-expressing images and used Hoechst stain images as a *post hoc* visual check on our fit only (*Figure 5*; *Figure 5—figure supplements 1–2*).

## Image processing

Images recovered from the time series above were processed into binary ('infectious' vs. 'non-infectious' or, for Hoechst-stained images, 'live' vs. 'dead') form using the EBImage package (*Pau et al., 2010*) in R version 3.6 for MacIntosh, after methods further detailed in *Supplementary file 7*. Binary images were then further processed into time series of infectious or, for Hoechst-stained images, live cells using a series of cell counting scripts. Because of logistical constraints (i.e. many plates of simultaneously running infection trials and only one available imaging microscope), the time course of imaging across the duration of each trial was quite variable. As such, we fitted a series of statistical models to our processed image data to reconstruct reliable values of the infectious proportion of each well per hour for each distinct trial in all cell line–virus-MOI combinations (*Figure 1—figure supplements 2–3*), as well as for declining live cell counts from control well data derived from the Hoestch time series (*Supplementary file 1*; *Supplementary file 7*; *Figure 1—figure supplement 7*). All original and processed images, image processing and counting code, and resulting time series data are freely available for download at the following FigShare repository: DOI: 10.6084/m9.figshare.8312807.

## **Mean field model**

### Theoretical model details

To derive the expression for $R_0$, the basic pathogen reproductive number in vitro, we used Next Generation Matrix (NGM) techniques (*Diekmann et al., 1990*; *Heffernan et al., 2005*), employing Wolfram Mathematica (version 11.2) as an analytical tool. $R_0$ describes the number of new infections generated by an existing infection in a completely susceptible host population; a pathogen will invade a population when $R_0>1$ (*Supplementary file 2*). We then analyzed stability properties of the system, exploring dynamics across a range of parameter spaces, using MatCont (version 2.2) (*Dhooge et al., 2008*) for Matlab (version R2018a) (*Supplementary file 3*).

### Theoretical model fitting

The birth rate, *b,* and natural mortality rate, μ, balance to yield a population-level growth rate, such that it is impossible to estimate both *b* and μ simultaneously from total population size data alone. As such, we fixed *b* at. 025 and estimated μ by fitting an infection-absent version of our mean field model to the susceptible time series derived via Hoechst staining of control wells for each of the three cell lines (*Figure 1—figure supplement 7*). This yielded a natural mortality rate, μ, corresponding to a lifespan of approximately 121, 191, and 84 hours, respectively, for Vero, RoNi/7.1, and PaKiT01 cell lines (*Figure 1—figure supplement 7*). We then fixed the virus incubation rate, σ, as the inverse of the shortest observed duration of time from initial infection to the observation of the first infectious cells via fluorescent microscope for all nine cell line – virus combinations (ranging 6 to 9.5 hours). We fixed α, the infection-induced mortality rate, at 1/6, an accepted standard for general viral kinetics (*Howat et al., 2006*), and held *c,* the rate of antiviral cell regression to susceptible status, at 0 for the timespan (<200 hours) of the experimental cell line infection trials.

We estimated cell line–virus-MOI-specific values for β, ρ, and ε by fitting the deterministic output of infectious proportions in our mean field model to the full suite of statistical outputs of all trials for

each infected cell culture time series (*Figure 1—figure supplements 2–3*). Fitting was performed by minimizing the sum of squared differences between the deterministic model output and cell line–virus-MOI-specific infectious proportion of the data at each timestep. We optimized parameters for MOI = 0.001 and 0.0001 simultaneously to leverage statistical power across the two datasets, estimating a different transmission rate, β, for trials run at each infectious dose but, where applicable, estimating the same rates of ρ and $\varepsilon$ across the two time series. We used the differential equation solver lsoda() in the R package deSolve (*Soetaert et al., 2010*) to obtain numerical solutions for the mean field model and carried out minimization using the 'Nelder-Mead' algorithm of the optim() function in base R. All model fits were conducted using consistent starting guesses for the parameters, β (β = 3), and where applicable, ρ (ρ = 0.001) and $\varepsilon$ ($\varepsilon$ = 0.001). In the case of failed fits or indefinite hessians, we generated a series of random guesses around the starting conditions and continued estimation until successful fits were achieved.

All eighteen cell line–virus-MOI combinations of data were fit by an immune absent ($\varepsilon$ = ρ = 0) version of the theoretical model and, subsequently, an induced immunity ($\varepsilon$ = 0; ρ >0) and constitutive immunity ($\varepsilon$ >0; ρ >0) version of the model. Finally, we compared fits across each cell line–virus-MOI combination via AIC. In calculating AIC, the number of fitted parameters in each model (*k*) varied across the immune phenotypes, with one parameter (β) estimated for absent immune assumptions, two (β and ρ) for induced immune assumptions, and three (β, ρ, and $\varepsilon$) for constitutive immune assumptions. The sample size (*n*) corresponded to the number of discrete time steps across all empirical infectious trials to which the model was fitted for each cell-line virus combination. All fitting and model comparison scripts are freely available for download at the following FigShare repository: DOI: 10.6084/m9.figshare.8312807.

## Spatial model simulations

Finally, we verified all mean field fits in a spatial context, in order to more thoroughly elucidate the role of antiviral cells in each time series. We constructed our spatial model in C++ implemented in R using the packages Rcpp and RcppArmadillo (*Eddelbuettel and Francois, 2011*; *Eddelbuettel and Sanderson, 2017*). Following *Nagai and Honda (2001)* and *Howat et al. (2006)*, we modeled this system on a two-dimensional hexagonal lattice, using a ten-minute epidemic timestep for cell state transitions. At the initialization of each simulation, we randomly assigned a duration of natural lifespan, incubation period, infectivity period, and time from antiviral to susceptible status to all cells in a theoretical monolayer. Parameter durations were drawn from a normal distribution centered at the inverse of the respective fixed rates of μ, σ, α, and c, as reported with our mean field model. Transitions involving the induced (ρ) and constitutive ($\varepsilon$) rates of antiviral acquisition were governed probabilistically and adjusted dynamically at each timestep based on the global environment. As such, we fixed these parameters at the same values estimated in the mean field model, and multiplied both ρ and $\varepsilon$ by the global proportion of, respectively, exposed and susceptible cells at a given timestep.

In contrast to antiviral acquisition rates, transitions involving the birth rate (*b*) and the transmission rate (β) occurred probabilistically based on each cell's local environment. The birth rate, *b*, was multiplied by the proportion of susceptible cells within a six-neighbor circumference of a focal dead cell, while β was multiplied by the proportion of infectious cells within a thirty-six neighbor vicinity of a focal susceptible cell, thus allowing viral transmission to extend beyond the immediate nearest-neighbor boundaries of an infectious cell. To compensate for higher thresholds to cellular persistence and virus invasion which occur under local spatial conditions (*Webb et al., 2007*), we increased the birth rate, *b*, and the cell-to-cell transmission rate, β, respectively, to six and ten times the values used in the mean field model (*Supplementary file 4*). We derived these increases based on the assumption that births took place exclusively based on pairwise nearest-neighbor interactions (the six immediately adjacent cells to a focal dead cell), while viral transmission was locally concentrated but included a small (7.5%) global contribution, representing the thirty-six cell surrounding vicinity of a focal susceptible. We justify these increases and derive their origins further in *Supplementary file 5*.

We simulated ten stochastic spatial time series for all cell-virus combinations under all three immune assumptions at a population size of 10,000 cells and compared model output with data in *Figure 5—figure supplement 3*. Spatial model code is available for public access at the following FigShare repository: DOI: 10.6084/m9.figshare.8312807.

## Acknowledgements

CEB was supported by a National Science Foundation Graduate Research Fellowship at Princeton University, a Miller Institute for Basic Research Fellowship at UC Berkeley, a DARPA PREEMPT program Cooperative Agreement grant D18AC00031, and an NIH grant 1R01AI129822-01. Tissue culture experiments were funded by an NIH grant R01 AI134824 to KC. Work in LFW's lab was funded by the Singapore National Research Foundation grants (NRF2012NRF-CRP001-056 and NRF2016NRF-NSFC002-013). CD was supported by the German Research Council (DFG) grant DFG SPP1596 (DR 772/10–2), the Federal Ministry of Education and Research (BMBF) grant RAPID (#01KI1723A) and the EU Horizon 2020 grant EVAg (#653316). The authors thank the Chandran lab at Albert Einstein College of Medicine – in particular, Cecilia Harold, Megan Slough, Rohit Jangra, and Tanwee Alkutkar – for technical support during tissue culture experiments. The authors further thank Jessica Metcalf and the Graham lab at Princeton for conceptual guidance throughout the project's development.

## Additional information

### Funding

| Funder | Grant reference number | Author |
|---|---|---|
| National Science Foundation | Graduate Research Fellowship | Cara E Brook |
| Adolph C. and Mary Sprague Miller Institute for Basic Research in Science, University of California Berkeley | Postdoctoral Fellowship | Cara E Brook |
| National Institutes of Health | R01-AI134824 | Kartik Chandran |
| Singapore National Research Foundation | NRF2012NRF-CRP001-056 | Lin-Fa Wang |
| Singapore National Research Foundation | NRF2016NRF-NSFC002-013 | Lin-Fa Wang |
| Deutsche Forschungsgemeinschaft | DR 772/10-2 | Christian Drosten |
| Bundesministerium für Bildung und Forschung | RAPID #01KI1723A | Christian Drosten |
| Horizon 2020 | #653316 | Christian Drosten |
| DARPA | PREEMPT Program - Cooperative Agreement no. D18AC00031 | Cara E Brook |
| National Institutes of Health | 1R01AI129822-01 | Cara E Brook |

The funders had no role in study design, data collection and interpretation, or the decision to submit the work for publication.

### Author contributions

Cara E Brook, Conceptualization, Data curation, Formal analysis, Investigation, Visualization, Methodology; Mike Boots, Andrew P Dobson, Conceptualization, Supervision; Kartik Chandran, Resources, Supervision, Funding acquisition; Christian Drosten, Marcel A Müller, Lin-Fa Wang, Resources, Methodology; Andrea L Graham, Bryan T Grenfell, Conceptualization, Supervision, Investigation; Melinda Ng, Conceptualization, Data curation, Validation, Investigation, Methodology; Anieke van Leeuwen, Conceptualization, Formal analysis, Supervision, Investigation, Visualization

### Author ORCIDs

Cara E Brook https://orcid.org/0000-0003-4276-073X
Kartik Chandran http://orcid.org/0000-0003-0232-7077
Andrea L Graham http://orcid.org/0000-0002-6580-2755

Marcel A Müller [ID] http://orcid.org/0000-0003-2242-5117
Lin-Fa Wang [ID] http://orcid.org/0000-0003-2752-0535
Anieke van Leeuwen [ID] https://orcid.org/0000-0003-1987-1458

## Decision letter and Author response
Decision letter https://doi.org/10.7554/eLife.48401.sa1
Author response https://doi.org/10.7554/eLife.48401.sa2

## Additional files

### Supplementary files

• Supplementary file 1. (A) Raw proportion infectious from cell culture images. Dataset gives raw proportion of infectious cells and time elapsed for all trials of all cell line/virus/MOI combinations, derived from image processing of binary images. (B) Raw proportion uninfectious from cell culture images. Dataset gives raw proportion of uninfectious cells and time elapsed for all trials of all cell line/virus/MOI combinations, derived from image processing of binary Hoechst-stained images. (C) Statistical mean of infectious time series for all trials of each cell line/virus/MOI experiment, from GAM fitted model incorporating random effects by trial. Data were smoothed to yield the proportion infectious per hourly timestep for each trial, and mean field mechanistic models were fit to the smoothed mean of all compiled trials for each cell line/virus/MOI combination. (D) Statistical mean of uninfectious time series for all eighteen cell line/virus/MOI experiments, from generalized linear model fit to Hoechst stain data reported on tab B. Note that these means were not used in epidemic model fitting but natural mortality rates for each cell line were derived from fitting an infection-absent model to the trajectory of susceptible decline for control trials for each cell line, as shown in *Figure 1—figure supplement 7*. All original raw image files, processed binary images, and image processing code are available freely for download at the following FigShare repository: DOI: 10.6084/m9.figshare.8312807.

• Supplementary file 2. Derivation of $R_0$.

• Supplementary file 3. Special points from bifurcation analysis.

• Supplementary file 4. Optimized parameters from all deterministic model outputs and spatial approximations.

• Supplementary file 5. Justification for parameter increase from mean field to spatial model.

• Supplementary file 6. Primers for qPCR.

• Supplementary file 7. Detailed methods for image and image data processing.

• Transparent reporting form

### Data availability

All data generated or analysed during this study are included in the manuscript and supporting files. All images and code used in this study have been made available for download at the following Figshare repository: https://doi.org/10.6084/m9.figshare.8312807.

The following dataset was generated:

| Author(s) | Year | Dataset title | Dataset URL | Database and Identifier |
|---|---|---|---|---|
| Brook CE, Ng M, Boots M, Dobson A, Graham A, Grenfell B, Chandran KC, van Leeuwen A | 2019 | Data and Code from: Accelerated viral dynamics in bat cell lines, with implications for zoonotic emergence | https://doi.org/10.6084/m9.figshare.8312807 | figshare, 10.6084/m9.figshare.8312807 |

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
