## [Decision Letter]

**Acceptance summary:**

This paper is an excellent example of how in vitro models of cell-virus interactions can be used to shape and formulate more general and larger-scale hypotheses about epidemiological dynamics. In this case, the choice of bat cell lines expressing induced and constitutive immune phenotypes enables estimates of different viral propagation rates. The results suggest that if bat cells do have greater constitutive immunity, this could lead to situations in which viruses that do propagate in bats will do so with much greater vigour (and possibly virulence and transmissability) should they 'spill-over' into non-bat hosts. The paper should be of wide general interest to those with interests in emerging disease dynamics and to quantitative biologists interested in the mathematical modelling of in vitro systems.

**Decision letter after peer review:**

Thank you for submitting your article "Within-host dynamics of virulent viruses in bat reservoirs for emerging zoonotic disease" for consideration by *eLife*. Your article has been reviewed by three peer reviewers, including Dan Haydon as the Reviewing Editor and Reviewer #3, and the evaluation has been overseen by Neil Ferguson as the Senior Editor.

The reviewers have discussed the reviews with one another and the Reviewing Editor has drafted this decision to help you prepare a revised submission.

Summary:

There is a view that bats (both as individuals and populations) are able to maintain a large number of virus species (often with little or no pathology) that are highly pathogenic outside of this order, and this paper seeks mechanistic reasons for why this might be. Specifically the authors set out to explore whether this might be due to uniquely constitutive immune capabilities of bat cells. To explore this hypothesis, the authors study the dynamics of 3 strains of vesicular stomatitis virus that express different sorts of glycoprotein, in 3 different cell cultures: Vero (non-bat cells with limited antiviral capabilities); Egyptian Fruit Bat cells which demonstrate an idiosyncratic induced interferon response, and Black Fruit Bat cells which constitutively express interferon. Well-mixed and imperfectly-mixed frequency dependent models of the infection dynamics are fitted to the means of these replicated time-series, and the models constructed to reflect the possibly differential anti-viral capabilities of the cell lines. The parameter estimates from the model fitting process lead to the key conclusion that 'induced immune responses favor slower cell-to-cell transmission rates (a proxy for viral replication rates) […] while constitutive immunity amplifies cellular transmission rates, in conjunction with less rapid antiviral responses.' The authors further conclude that '[i]f hosts with constitutive immune defenses favor evolution of rapidly transmitting viruses, such pathogens are likely to cause extreme virulence in spillover hosts lacking similarly constitutive defenses'.

One reviewer indicates that if the paper has set out to answer the question as to whether the immune phenotype of bats alters the capacity for viruses to persist within them, the answers are not sufficiently clear to merit publication in *eLife*. This is an understandable point of view: it seems that none of the best-fitting well-mixed models are constitutive, and only one of the spatially explicit models is constitutive where we might expect it to be (and one where we wouldn't expect it to fit best). However, I have decided to go with the majority view and provide an opportunity for revision, but the authors should consider carefully whether the essential points below can be met (or convincingly rebutted).

Essential revisions:

1) Overall, the reviewers felt that this paper was quite a bit more complicated in the presentation of its results than it needs to be, and I encourage the authors to find ways of simplifying the Results section and linearizing the framing of the key messages. I wondered what the benefits of presenting both the well-mixed and spatial models were? Which is the most appropriate? Do they say importantly different things? Could the manuscript be simplified by focusing on just one?

2) Another key question the authors should consider is whether the data in Figure 4 really support (or even need) the cell turnover embedded within the formulation of the model? Could the main point of the paper be more clearly made by fitting simpler models more consistent with the short time frame of the experimental data, and concentrating on the initial infection spread and sometime declines, rather than a scenario requiring cell turnover? If the cell cultures were capable of maintaining infection over the longer term, why weren't the experiments run over longer time frames to demonstrate this?

Introduction:

3) The authors cannot claim to be studying the within-host dynamics of bat viruses. Rather, they experimentally examine and model the dynamics of recombinant viruses in vitro. The authors should explicitly recognize throughout their manuscript that their study is at this more limited scale and, as a result, that its impact is also limited.

Results/Materials and methods:

4) The three cell lines are presented as examples of no immunity, induced immunity and constitutive immunity. The questions set out at the end of the Introduction are not answered directly (for example at the beginning of the Discussion) with attention instead diverted to measures of rate of spread. Ensure a more identifiable match between the questions posed at the end of the Introduction, and the start of the Discussion.

5) Add more explanation of what these virus lines are and why they are chosen. The reader is not told that rVSV-MARV and rVSV-EBOV are recombinant viruses expressing Ebola and Marburg Virus (Results first paragraph). Moreover, it is not stated why the authors employ two viruses (they do not seem to be interested in contrasting them but they cannot be said to be true replicates) or these two in particular.

6) How many replicates were there for each cell-line/clone combination? The only detail mentioned in this regard is that there were 2-3 technical replicates per plate. Such details should be stated prominently in the Results, since large numbers of points are plotted, but the reader has no way to discern which of them are independent.

7) There is a strong case for requiring the model is fitted to the individual replicates and not the mean of the replicates. If this case is rejected, then careful justification must be provided.

8) The cell cultures are grown to 90% confluency, and are modeled as systems in which there is cell birth and natural death (birth rate about 4x the natural death rate). Readers are likely to be left with a number of questions that the authors should address more explicitly: Where does the birth rate parameter come from? Is this a realistic estimate reflective of the dynamics of the cell culture, or is it a deduction based on the requirement to sustain the infection dynamics (either endemic or oscillatory?). The Materials and methods suggest that it is what would be required to maintain sustainable live cell populations, but is this what actually happens in the culture during the infection? What sort of turnover does this result in, in the absence of infection? What is the justification for frequency over density dependent dynamics? At the heart of these questions is whether it is appropriate to think of these cell cultures as a sort of viral chemostat which could maintain steady state or oscillatory dynamics indefinitely, or whether the system would be better modeled as an infection process in a fixed non-reproducing host population. The latter view would substantially simplify the paper if such a perspective was justifiable.

9) The potential influence of the assumed value for the rate at which antiviral cells regress to susceptibility is unclear. At present, it is not clear where the parameter value used comes from and what impact it could have on the results. This is particularly pertinent since sustained epidemics are ascribed to the antiviral cells returning to susceptibility – can one confidently separate this from birth of new susceptibles?

10) Why is ε not estimated? It seems to be fixed at 0 or 1. In subsection “Fitting of theoretical model to cell culture data demonstrates higher within-host transmission rates under constitutive immune assumptions” this seems to be recognized but Supplementary file 4 doesn't provide any intermediate values. It wasn't clear to why ε is bounded at 1. It is a rate?

11) It is surely odd that as b (and u) is lowered towards zero, R_0_ also tends to zero. Surely one would expect a virus to be able to spread in a non-renewing monolayer?

Discussion:

12) The use of the word 'favor' in the first paragraph of the Discussion is a little misleading and should be re-worded. These findings are not based on any evolutionary process.

13) The Discussion should make clear that the question of whether bats in general have constitutively active intracellular immunity is far from resolved. As touched on in the Introduction, this partly depends on definition – does constitutive expression of e.g. interferon-α actually provide constitutive protection? What about other immune genes which seem to be defective (e.g. Xie et al., 2018)? More generally, the immune systems of only a very small fraction of bat diversity (and of non-lab animals in general) has been studied, so it remains unclear how well we can generalise to all bats, or whether bats are truly unusual relative to other wild animals. Where possible the authors should list examples of studies showing constitutive immunity specifically in the species (or close relatives of these) thought to be reservoirs for the highly virulent viruses discussed.

---

## [Author Response]

Essential revisions:1) Overall, the reviewers felt that this paper was quite a bit more complicated in the presentation of its results than it needs to be, and I encourage the authors to find ways of simplifying the Results section and linearizing the framing of the key messages. I wondered what the benefits of presenting both the well-mixed and spatial models were? Which is the most appropriate? Do they say importantly different things? Could the manuscript be simplified by focusing on just one?

Thank you for taking the time to decipher our paper, and we apologize for any unnecessary complexity and confusing organization and/or writing. As highlighted above, our new version of the manuscript presents a completely reworked version of the model and, correspondingly, a complete rewrite of the text. The main messages of our paper are threefold: (1) we find support for immune phenotypes suggested from the literature for each of the three cell lines, in particular showing a necessary effect of top-down immune response in describing data for the bat cell lines (especially the constitutively antiviral PaKiT01 cell line); (2) we find that more pronounced immune responses (either induced or constitutive) are correlated with higher cell-to-cell viral transmission rates (β), suggesting that bats’ robust antiviral immune phenotype could be driving the evolution of higher within-host viral replication rates likely to cause virulence upon emergence into hosts lacking bat immune capacities; and (3) IFN-induced antiviral cells promote the maintenance of persistent within-host infections by protecting live cell refugia against rapid viral infection and mortality in the initial epidemic takeoff such that live cells are maintained that can sustain transmission at later timepoints.

Key differences in the new version of the model which depart from the original model include (a) no regression of antiviral cells back to susceptible status and (b) estimation of the rate of constitutive cellular acquisition of antiviral status (represented as ε), rather than fixation as a 0-1 scalar as in the first version of submission. We elaborate on the effects of these major changes to the model in answer to questions #2, 9, and 10 below.

Finally, in response to your questions above, we have largely relegated the spatial model to the supplementary materials and are here using it only to validate and visualize results from our mean field model fitting. The mean field model demonstrates all of our key takeaway messages from this manuscript, with the added benefit of being generalizable to the whole organism when considered theoretically in our bifurcation analysis. However, since our data (in plaque assay) are spatial, we felt that it was important to simulate the dynamics in a spatial context as well. We find that, in doing this, our key takeaways are reaffirmed and validated. We retain the spatial stochastic videos in our supplementary text because they clearly illustrate how the epidemic wave ‘washes over’ antiviral refugia, which can become infected at later timepoints to sustain transmission. This pattern is seen in the mean field model as well (Figure 5, main text, and Figure 5—figure supplements 1 and 2) but much easier to interpret when visualized spatially.

2) Another key question the authors should consider is whether the data in Figure 4 really support (or even need) the cell turnover embedded within the formulation of the model? Could the main point of the paper be more clearly made by fitting simpler models more consistent with the short time frame of the experimental data, and concentrating on the initial infection spread and sometime declines, rather than a scenario requiring cell turnover? If the cell cultures were capable of maintaining infection over the longer term, why weren't the experiments run over longer time frames to demonstrate this?

In the newest version of the manuscript, the original Figure 4 is now Figure 1, showing the fitted infectious time series to these data. In response to this comment, we attempted to fit several versions of the model that incorporated only the natural mortality rate (fitted for each cell line) and the epidemic dynamics of each time series, with no birth rate included in the model. However, these versions of the model were unable to re-capture the extended time series witnessed in the rVSV-EBOV infections on PaKiT01 cell lines, using fitted mortality (μ) and incubation (σ) rates and a fixed, standard infection-induced mortality rate (α= 1/6). Thus, cell turnover was needed in order to recapture the data.

Additionally, we feel confident that cell turnover *did* take place in the experiments because we grew cells to 90% confluency prior to initiating the plaque assay, then watched them subsequently fill the plate when they could not be split at later timepoints. In including cell turnover, our analysis additionally maintains consistency with previous in vitromodeling of IFN dynamics in cell culture in the general literature (Howat et al., 2006).

Introduction:3) The authors cannot claim to be studying the within-host dynamics of bat viruses. Rather, they experimentally examine and model the dynamics of recombinant viruses in vitro. The authors should explicitly recognize throughout their manuscript that their study is at this more limited scale and, as a result, that its impact is also limited.

Fair point, and thank you for bringing up this issue. We do believe that our mean field model can be generalized to the level of the whole host organism, as a description of viral spread throughout a host’s tissue, which is why we have undertaken the broad bifurcation analysis, in addition to fitting our model to the in vitrodata. Figure 2 and 3, as well as the derivation for R_0_ (as a threshold for tissue invasion) presented in our text are, thus, truly representative of within-host dynamics. Nonetheless, we appreciate your concerns and have scaled back our claims to highlight the terms ‘in vitro*’* or ‘bat cell culture’ where appropriate, including in the title.

Results/Materials and methods:4) The three cell lines are presented as examples of no immunity, induced immunity and constitutive immunity. The questions set out at the end of the Introduction are not answered directly (for example at the beginning of the Discussion) with attention instead diverted to measures of rate of spread. Ensure a more identifiable match between the questions posed at the end of the Introduction, and the start of the Discussion.

Thanks for this! Good point, and apologies for any digressions in our writing. As mentioned above, our newest version of the manuscript is a complete rewrite of the original, and we have addressed this discrepancy. The end of our Introduction now highlights our main hypotheses:

“(a) We hypothesized that model fitting would demonstrate the most significant role for top-down immune processes in bat cell lines described as constitutively antiviral in the literature. We further predicted that the most robust antiviral responses would (b) be correlated with the most rapid within-host virus propagation rates while (c) protecting cells against virus-induced mortality to support the longest enduring infections in tissue culture.”

And the beginning of our Discussion addresses these hypotheses:

“Critically, we found that – as hypothesized – (a) bat cell lines demonstrated a signature of enhanced interferon-mediated immune response, of either constitutive or induced form, which (b) allowed for establishment of rapid within-host, cell-to-cell virus transmission rates (β). These results were supported by both data-independent bifurcation analysis of our mean field theoretical model, as well as fitting of this model to viral infection time series established in bat cell culture. Additionally, we (c) demonstrated how the antiviral state induced by the interferon pathway protects live cells from mortality in tissue culture, resulting in in vitro epidemics of extended duration and an enhanced probability of establishing a long-term persistent infection.”

5) Add more explanation of what these virus lines are and why they are chosen. The reader is not told that rVSV-MARV and rVSV-EBOV are recombinant viruses expressing Ebola and Marburg Virus (Results first paragraph). Moreover, it is not stated why the authors employ two viruses (they do not seem to be interested in contrasting them but they cannot be said to be true replicates) or these two in particular.

Thank you for bringing this up. We’ve added to our Viruses section in the Materials and methods portion of the manuscript to further elaborate on and explain why the viruses were selected, as well as extended our Introduction to these time series upon first presentation in the Results section. We included both rVSV-EBOV and rVSV-MARV because we wanted to include a known constitutively antiviral cell line-virus combination (rVSV-MARV on PaKiT01 cells), over and above the constitutive expression of IFN-α in these cells.

6) How many replicates were there for each cell-line/clone combination? The only detail mentioned in this regard is that there were 2-3 technical replicates per plate. Such details should be stated prominently in the Results, since large numbers of points are plotted, but the reader has no way to discern which of them are independent.

We now highlight the number of replicates (18-39) in the Results section and the Materials and methods. Additionally, we include dashed statistical fits to each trial (see response to #7 below) in two figure supplements (Figure 1—figure supplement 2 and Figure 1—figure supplement 3) to further differentiate each time series.

7) There is a strong case for requiring the model is fitted to the individual replicates and not the mean of the replicates. If this case is rejected, then careful justification must be provided.

Thank you, and yes, we agree. In our newest version of the manuscript, we have refitted the mean field model to all individual replicates for each cell line-virus-MOI combination, rather than to the mean of replicates. We find our fits to be much improved and now consistent with our hypotheses.

8) The cell cultures are grown to 90% confluency, and are modeled as systems in which there is cell birth and natural death (birth rate about 4x the natural death rate). Readers are likely to be left with a number of questions that the authors should address more explicitly: Where does the birth rate parameter come from? Is this a realistic estimate reflective of the dynamics of the cell culture, or is it a deduction based on the requirement to sustain the infection dynamics (either endemic or oscillatory?). The Materials and methods suggest that it is what would be required to maintain sustainable live cell populations, but is this what actually happens in the culture during the infection? What sort of turnover does this result in, in the absence of infection? What is the justification for frequency over density dependent dynamics? At the heart of these questions is whether it is appropriate to think of these cell cultures as a sort of viral chemostat which could maintain steady state or oscillatory dynamics indefinitely, or whether the system would be better modeled as an infection process in a fixed non-reproducing host population. The latter view would substantially simplify the paper if such a perspective was justifiable.

Thanks for these valid and important insights and questions. We touched a bit on these questions in our responses to point #2 above, but we will elaborate further here. We did attempt to fit several versions of the model that incorporated only the natural mortality rate (fitted for each cell line) and the epidemic dynamics of each time series, with no birth rate included in the model. These versions of the model were unable to re-capture the extended time series witnessed in the rVSV-EBOV infections on PaKiT01 cell lines, using fitted mortality (μ) and incubation (σ) rates and a fixed, standard infection-induced mortality rate (α= 1/6). Thus, cell turnover was needed in order to recapture the data, or we would need to re-evaluate the infection-induced mortality parameter. In order to recapture our data, however, the duration of the infectious period this parameter would necessarily need to be extended to allow for cells to produce infectious virus for over a week without dying in order to recover the sustained 200 hour infectious time series witnessed in our data. This is very unlikely.

In our most recent version of the manuscript, in order to address the criticisms highlighted above, we chose to fit a unique natural mortality rate (μ) for each cell line, rather than fixing all μ at a constant and consistent value as done previously. In order to do this (as highlighted in the Figure 1—figure supplement 7) and discussed in the Materials and methods), we fixed the birth rate, *b,* at.025, then fit a μ for each cell line to recover the statistical mean of the declining susceptible count of all control well trials for that cell line. The net effect of these birth-death fits is still cell loss in tissue culture over time but with some cell turnover taking place. We feel confident that cell turnover *did* take place because we grew cells to 90% confluency prior to initiating the plaque assay, then watched them subsequently fill the plate when they could not be split at later timepoints. In including cell turnover, our analysis additionally maintains consistency with previous in vitromodeling of IFN dynamics in cell culture in the general literature (Howat et al., 2006).

The reviewers additionally questioned our decision to model these dynamics in a frequency vs. density-dependent context. We feel this assumption was well supported in the current context because plaque assays restrict transmission neighbor-to-neighbor in two dimensional space, thus limiting the number of possible infectious contacts for a single cell at all timesteps, in keeping with frequency-dependent dynamics. Intriguingly, because dead cells occupy space in our model, however, this has the effect of reducing the transmission rate in a density-dependent fashion: declines in susceptible and infectious cell density are realized dynamically as declines in their proportions where dead cells fill their place and do not contribute to epidemic dynamics. This effect makes the mean field model more generalizable to the level of the entire organism.

9) The potential influence of the assumed value for the rate at which antiviral cells regress to susceptibility is unclear. At present, it is not clear where the parameter value used comes from and what impact it could have on the results. This is particularly pertinent since sustained epidemics are ascribed to the antiviral cells returning to susceptibility – can one confidently separate this from birth of new susceptibles?

Agreed, and thank you for highlighting this potential point of controversy. Certainly, antiviral cells do have the capacity to regress to susceptible status (see Samuel and Knutson, 1982; Rasmussen and Farley, 1975; Radke et al., 1974), but after reconsideration of our assumptions and more intensive review of the literature, we agree with this reviewer critique that this is unlikely to occur on the 200 hour timescale of our tissue cultures, especially in such a limited spatial environment with live virus still present (and therefore IFN-inducing) in the tissue culture. As a result of this, we have fixed the value of *c* at 0 in all model fits (and bifurcation analyses) in the newest version of the manuscript, such that the regeneration of susceptibles to maintain late stage epidemic transmission can now only occur as a result of susceptible cell births. This greatly simplifies our model and also produces more robust model fits to the data. We have maintained the term *c* in our model equations because this antiviral regression phenomenon is relevant at the scale of the whole organism, and we wished to keep our derivations for R_0_ generalizable.

10) Why is ε not estimated? It seems to be fixed at 0 or 1. In subsection “Fitting of theoretical model to cell culture data demonstrates higher within-host transmission rates under constitutive immune assumptions” this seems to be recognized but Supplementary file 4 doesn't provide any intermediate values. It wasn't clear to why ε is bounded at 1. It is a rate?

Previously, ε was modeled as a 0-1 scaler that modified the rate of antiviral acquisition (ρ), which is why it was fixed. However, your comments inspired us to re-structure the model to estimate the induced immune rate of antiviral acquisition (ρ), which is multiplied by the proportion of exposed cells in tissue culture, separately from the constitutive rate of antiviral acquisition (ε), such that – to answer your questions – (a) ε is now estimated under models assuming some degree of constitutive immunity (it is fixed at 0 for induced-only and absent immune models) and (b) yes, ε is a rate (though it was not previously). We find this a much simpler way to present the model, allowing for greater flexibility in parameter estimation and more accurate fits to the data.

11) It is surely odd that as b (and u) is lowered towards zero, R_0_ also tends to zero. Surely one would expect a virus to be able to spread in a non-renewing monolayer?

In a simple, density-dependent model with no recovery from infection, R_0_ will be written as βN/(α + μ), indicating that the pathogen is unable to spread at low population densities. Though our transmission rates are cell-to-cell frequency-dependent, our model is really density -dependent as a result of the inclusion of dead cells as a state variable and, therefore, has cell density thresholds for pathogen invasion. In the supplementary material of this manuscript, we derive the equation for Ps at disease free equilibrium, which is equal to (b-μ)(c+μ)b(c+μ+ε). This configuration of Ps is incorporated into our equation for R_0_ because a pathogen will not be able to invade a tissue with too high an initial proportion of dead cells (*P_D_*) relative to live cells (*P_S_*). Under conditions of no constitutive immunity (ε=0), this equation for *P_S_*reduces to (b-μ)b which then surfaces in the equation for R_0_. Because our system of differential equations was constructed including birth rates, the birth and death rates are incorporated in the formulation of the susceptible proportion at DFE. Thus, a monolayer with a higher death rate than birth rate will be composed of completely dead cells at DFE and, therefore, not be invadable by a pathogen.

We hope that this explanation provides some clarity. If easier to understand, we could rewrite R_0_ with the term *P_S_*instead of birth and death rates. Please let us know if you would prefer it this way.

Discussion:12) The use of the word 'favor' in the first paragraph of the Discussion is a little misleading and should be re-worded. These findings are not based on any evolutionary process.

Very true. Thanks for this. We’ve dropped the word wherever it was used in the text.

13) The Discussion should make clear that the question of whether bats in general have constitutively active intracellular immunity is far from resolved. As touched on in the Introduction, this partly depends on definition – does constitutive expression of e.g. interferon-α actually provide constitutive protection? What about other immune genes which seem to be defective (e.g. Xie et al., 2018)? More generally, the immune systems of only a very small fraction of bat diversity (and of non-lab animals in general) has been studied, so it remains unclear how well we can generalise to all bats, or whether bats are truly unusual relative to other wild animals. Where possible the authors should list examples of studies showing constitutive immunity specifically in the species (or close relatives of these) thought to be reservoirs for the highly virulent viruses discussed.

Great point – and thank you! We agree that this issue is far from resolved, and, in fact, we believe that our results largely support the opposite conclusion: that constitutive IFN expression functions more like a primed, inducible defense than a constitutive, functional antiviral protein. We have emphasized this debate in the Introduction as well as discussed it in the Results and in the Discussion.